# Policy Optimization in a Noisy Neighborhood:
# On Return Landscapes in Continuous Control

**Nate Rahn**[*]
Mila, McGill University

**Pierluca D'Oro**[*]
Mila, Université de Montréal

**Harley Wiltzer**
Mila, McGill University

**Pierre-Luc Bacon**
Mila, Université de Montréal

**Marc G. Bellemare**
Mila, McGill University

## Abstract

Deep reinforcement learning agents for continuous control are known to exhibit significant instability in their performance over time. In this work, we provide a fresh perspective on these behaviors by studying the return landscape: the mapping between a policy and a return. We find that popular algorithms traverse *noisy neighborhoods* of this landscape, in which a single update to the policy parameters leads to a wide range of returns. By taking a distributional view of these returns, we map the landscape, characterizing failure-prone regions of policy space and revealing a hidden dimension of policy quality. We show that the landscape exhibits surprising structure by finding simple paths in parameter space which improve the stability of a policy. To conclude, we develop a distribution-aware procedure which finds such paths, navigating away from noisy neighborhoods in order to improve the robustness of a policy. Taken together, our results provide new insight into the optimization, evaluation, and design of agents.

## 1 Introduction

It is well-documented that agents trained with deep reinforcement learning can exhibit substantial variations in performance – as measured by their episodic return. The problem is particularly acute in continuous control, where these variations make it difficult to compare the end product of different algorithms or implementations of the same algorithm [11, 20] or even reliably measure an agent's progress from episode to episode [9]. A recurring finding is that simply averaging the return produced by a set of policies may be insufficient for rigorous evaluation.

In this paper, we demonstrate that high-frequency discontinuities in the mapping from policy parameters $\boldsymbol{\theta}$ to the return $R(\boldsymbol{\theta})$ are an important cause of return variation. As a consequence of these discontinuities, a single gradient step or perturbation to the policy parameters often causes important changes in the return, even in settings where both the policy and the dynamics are deterministic. Because an agent's parameters constantly change during training and should be robust to minute parametric perturbations, we argue that the *distribution* of returns in the neighborhood of $\boldsymbol{\theta}$ is in fact a better representative of its performance, both from an evaluation and an optimization perspective.

**Noisy neighborhoods in the return landscape.** We call the *return landscape* the mapping from $\boldsymbol{\theta}$ to $R(\theta)$, our main object of study. We show that the return often varies substantially within the vicinity of any given $\boldsymbol{\theta}$, forming what we call a *noisy neighborhood* of $\boldsymbol{\theta}$. Based on this observation, we demonstrate the usefulness of studying the landscape through the distribution of returns obtained from small perturbations of $\boldsymbol{\theta}$. In the important case where these perturbations result from a single gradient step, we call the resulting object the *post-update return distribution*.

---

[*]Equal contribution. Correspondence to {`nathan.rahn,pierluca.doro`}@mila.quebec.

37th Conference on Neural Information Processing Systems (NeurIPS 2023).

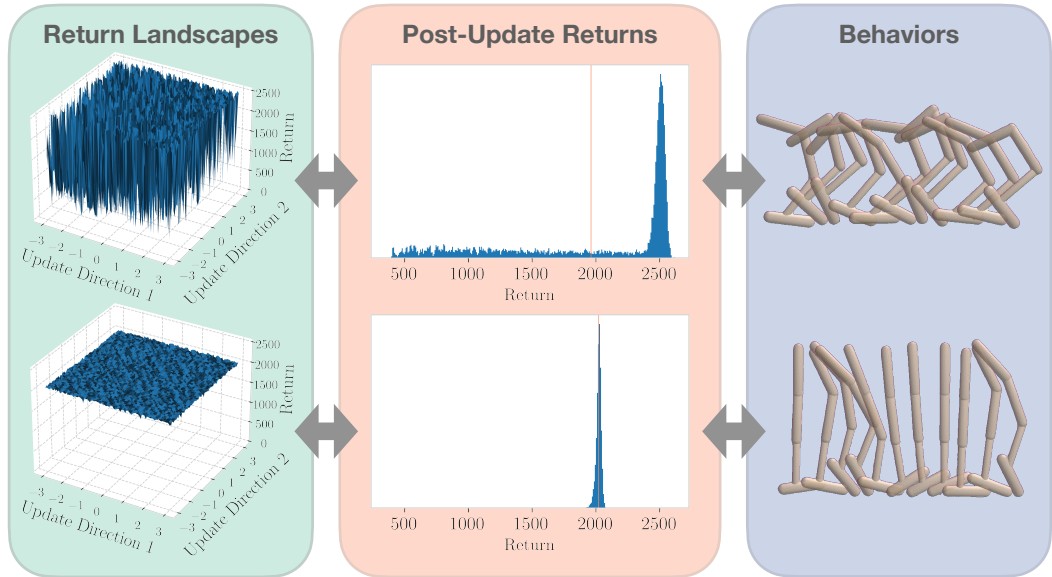

Figure 1: A visualization for two policies visited by SAC in the `hopper` environment. We show the return landscape in their proximity, their post-update return distributions, and the visual appearance of their learned gaits. We plot the mean of each return distribution as an orange line. Despite featuring a similar level of return, we observe that the policy in the noisy neighborhood performs an unstable curved gait which is faster but more prone to failure, as visible in the thick left tail of the post-update return distribution.

**Diversity in equally-performing policies.** We show that different neighborhoods correspond to different post-update return distributions and agent behaviors. We discover that at equal average returns, different policies obtained by the same deep RL algorithm may in fact have substantially different distributional profiles, as measured by statistics of the post-update return distribution. Moreover, we uncover that many of these distributions are long-tailed and we find the source of these tails to be sudden failures from an otherwise successful policy.

**Effect on learning dynamics.** We expose how the transition between noisy and smooth parts of the landscape happens. Surprisingly, while large valleys of low return are visible when linearly interpolating between similarly performing policies from different runs, we show no such valleys typically exist between policies from the same run. Based on this insight, we show that it is possible to find an improved set of parameters $\boldsymbol{\theta}$ that achieves comparable return, but substantially lower post-update variation.

We believe the phenomenon we study is central to deep reinforcement learning in continuous control. Beyond its effect on learning dynamics (for example, through increased variance and implicit exploration [39]), it is also a potential driver of instability in sim2real settings, even in the face of seemingly small environmental changes. Additionally, it suggests that one should not simply deploy the policy obtained at the end of a training run, and that further post-training tuning may be beneficial.

## 2 Background

In reinforcement learning, an agent interfaces with an environment. In this paper, we are interested in continuous control environments modelled as a finite-horizon Markov Decision Process (MDP) $\mathcal{M} = \langle \mathcal{S}, \mathcal{A}, r, f, T, \rho_0 \rangle$, where $\mathcal{S} \equiv \mathbb{R}^n$ is the state space, $\mathcal{A} \equiv \mathbb{R}^m$ is the action space, $r : \mathcal{S} \times \mathcal{A} \to \mathbb{R}$ is a reward function, $f : \mathcal{S} \times \mathcal{A} \to \mathcal{S}$ is a deterministic transition function, $T$ is the horizon, and $\rho_0 = \mathcal{U}(s_I - \beta, s_I + \beta)$ is an initial state distribution with $s_I$ being an initial reference state, and $\beta \in \mathbb{R}^n$ an environment-dependent parameter. We assume that each agent produces a stationary Markovian deterministic policy $\pi_{\boldsymbol{\theta}} : \mathcal{S} \to \mathcal{A}$ within a parametrized family $\Pi_{\Theta} = \{\pi_{\boldsymbol{\theta}} : \boldsymbol{\theta} \in \Theta \subset \mathbb{R}^d\}$. In an episodic setting, the interaction of the agent with the environment with a given policy $\pi_{\boldsymbol{\theta}}$ from some state $s \in \mathcal{S}$ produces a trajectory in the environment and,

consequently, a *return*:

$$G_{\boldsymbol{\theta}}(s) = \sum_{t=1}^{T} r(s_t, a_t) \tag{1}$$

$$\text{s.t. } s_t = f(s_{t-1}, a_{t-1}), a_t = \pi_{\boldsymbol{\theta}}(s_t), s_1 = s.$$

We are interested in understanding how small changes to the policy parameter affect the associated return. To this end it is sufficient to study the return from the reference state $s_I$ (in Appendix A.4 we show that similar effects occur across the state space). The *return landscape* is our main object of study.

**Definition 2.1** (Return Landscape). The return landscape is the mapping from policy parameters to return, starting from the initial reference state:

$$R(\boldsymbol{\theta}) = G_{\boldsymbol{\theta}}(s_I). \tag{2}$$

Figure 1 (left) depicts small portions of the return landscape for a particular environment and policy parametrization (we describe the visualization procedure below).

In this work, we will use the policies discovered by popular algorithms to characterize the topology of the return landscape. We focus on policy-based deep reinforcement learning algorithms for continuous control, such as Soft Actor-Critic (SAC) [19], Twin-Delayed DDPG (TD3) [16], and PPO [42] which use neural network function approximators to represent the policy. Such algorithms learn good behavior in the environment by maximizing the discounted return. In the process, they produce a sequence of policies

$$\boldsymbol{\theta}_0, \boldsymbol{\theta}_1, \ldots, \boldsymbol{\theta}_N, \text{ s.t. } \boldsymbol{\theta}_{t+1} = \mathfrak{u}(\boldsymbol{\theta}_t, X_t) \text{ for all } t, \tag{3}$$

where $\mathfrak{u} : \Theta \times \mathbb{R} \to \Theta$ is the algorithmic policy update function, and $X_t$ is some random variable abstracting the stochasticity inherent to the update. For example, SAC and TD3 construct parametric updates by sampling a small number of transitions (minibatches) from their replay buffer [29, 31].

## 3 A Distributional View on Return Landscapes

The return landscape arises from the interaction between an environment and a class of parameterized policies. We first consider how this landscape varies in the immediate vicinity (or neighborhood) of policies produced by deep reinforcement learning algorithms. Given a reference policy, a natural choice is to consider how the return is affected by single updates to the policy parameters. To this end, we view the collection of possible returns obtained by evaluating the updated policy as a distribution over returns; as we will see, this distribution widely varies across the return landscape.

**Definition 3.1** (Post-Update Return). Let $\Pi_\Theta$ be a parametric space of deterministic policies and $\mathfrak{u}$ an update function. Given $\pi_{\boldsymbol{\theta}} \in \Pi_\Theta$, its *post-update return* is defined as:

$$\mathcal{R}(\boldsymbol{\theta}) = R(\mathfrak{u}(\boldsymbol{\theta}, X)), \quad X \sim P, \tag{4}$$

where $P$ is a an algorithm-dependent source of stochasticity.

The post-update return inherits randomness from the underlying training algorithm and it is thus a random variable. Clearly, a post-update return will have an associated policy and trajectory, which are in turn random variables. In this work, we will leverage the distribution of post-update returns as a tool to investigate the properties of neighborhoods of the return landscape.

The different panels of Figure 1 illustrate how the return landscape in the neighborhood of $\boldsymbol{\theta}$ translates into different post-update return distributions. Here, the return landscape is visualized along two update directions computed by the training algorithm based on two different batches sampled from its replay buffer, such that 1.0 on each axis corresponds to a single parameter update in that direction (details in Appendix A.2). The middle panel shows the corresponding post-update return distribution estimated using 10000 samples. We find that the distribution from the noisy neighborhood (top) exhibits a significant left tail, while the distribution from the quieter neighborhood (lower) is concentrated around its mean. On the right, we illustrate the gait produced by the reference policies (the origin in the left panel). We find qualitatively that the policy in the noisy neighborhood exhibits a curved gait which is sometimes faster, but unstable, whereas the policy in the smooth neighborhood produces an upright gait which can be slower, yet is very stable. We include similar evidence for other environments in Appendix A.11.

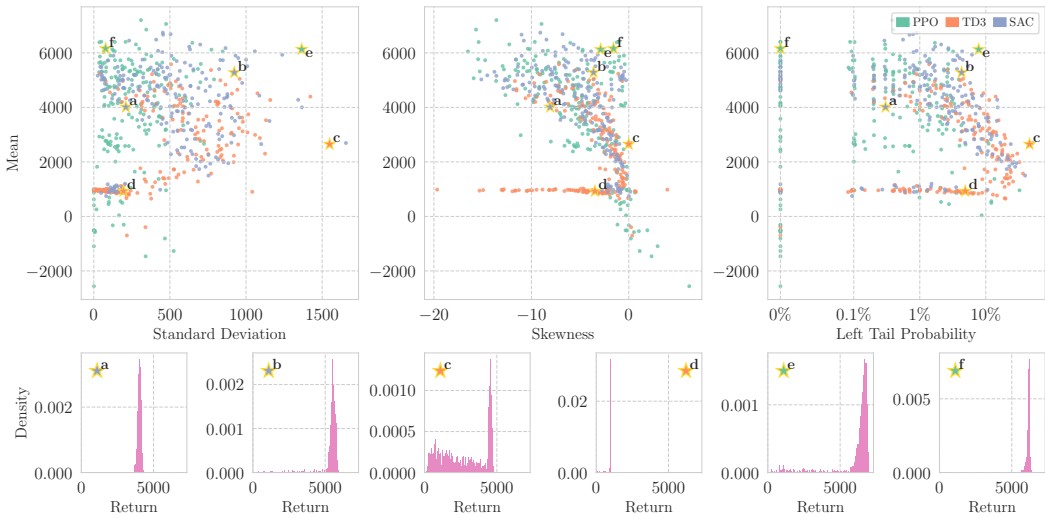

Figure 2: A scatter plot showing mean return and standard deviation, skewness or left-tail probability of the post-update return distribution of policies produced by three popular deep RL algorithms on the `ant` Brax task. Each point corresponds to a given policy's post-update return distribution, with six selected policies highlighted by star markers showing a range of diverse distributions.

### 3.1 Post-Update Return Distributions as a Characterization of the Return Landscape

The mean of the post-update distribution naturally captures the average behavior represented by an algorithm as it traverses a given neighborhood. We further characterize this distribution by measuring its standard deviation (a measure of spread around the mean) and its skewness (a measure of asymmetry). In our context, a negative skewness describes a distribution with a heavy left tail, similar to the one shown in Figure 1. Such a tail is especially interesting to us as it indicates lower-than-expected returns. However, we find that skewness is not directly interpretable as a numerical quantity. To capture these tails interpretably, we introduce a metric we call *left-tail probability*. The left-tail probability of a random variable $Y$ is defined as

$$\text{LTP}_\alpha(Y) = \mathbb{P}[0 \leq Y < \alpha \cdot \text{mode}(Y)]. \tag{5}$$

This quantity satisfies some desirable properties within the context of our study. First, it uses the mode of the distribution as a reference value. This is by contrast with the mean of the distribution, which may not correspond to the "majority" behavior (as illustrated in the top half of Figure 1). It also allows us to more easily compare the tailedness of distributions generated from policies of widely varying returns. Second, it is an easily-interpretable quantity which measures the total probability mass falling in the left tail. For simplicity, here we assume that $Y$ is positive, noting the idea can be naturally generalized to random variables bounded below. In our analyses we write $\text{LTP} \equiv \text{LTP}_{1/2}$ to measure drops from the mode of the post-update return distribution of at least 50%. In practice, we estimate the LTP by leveraging the Chernoff estimator [10], computing the mode as the midpoint of the interval of the most populated bin in a 100-bin histogram.

Equipped with these metrics, we measure the mean and the other three statistics of the post-update return for a set of 600 policies produced, across trials and iterations, by three popular deep RL algorithms (TD3, SAC and PPO). We use 20 seeds per algorithm and 10 checkpoints per seed, for a total of 200 policies per algorithm. These checkpoints are equally-spaced in time in training runs of 1 million steps for TD3 and SAC and 60 million steps for PPO. Each of the 600 distributions is estimated by performing 1000 independent updates to the starting policy and then rolling the resulting deterministic policy out in the environment for 1000 time-steps to compute its return. Each update is different due to a different batch sampled from the replay buffer for TD3 and SAC, and to a different batch of data from the environment collected by a randomly-perturbed policy for PPO. This amounts to millions of policy evaluations for which, for computational reasons, we primarily use the easily parallelizable environments from the Brax simulator [15]. We also include similar results on the post-update return distributions of policies trained on DeepMind Control Suite [44] and on games from the

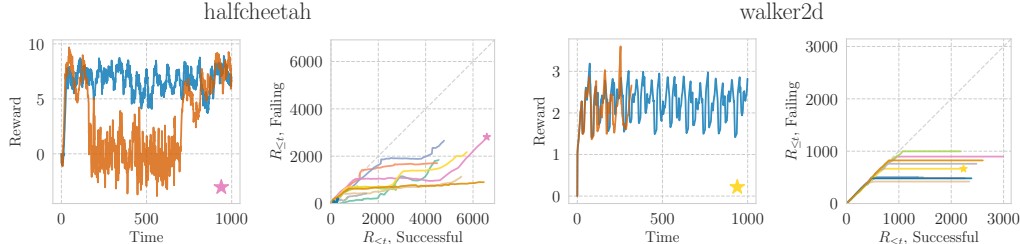

Figure 3: A visualization of how failures occur in the `halfcheetah` and `walker2d` tasks. The left subplots compare the reward-per-timestep obtained by a successful and failing trajectory generated by two policies in the same noisy neighborhood. The right subplots show the simultaneous evolution of returns for 10 such trajectory pairs (that can be thought of as a race to collect the most rewards), with the trajectory pair from the left indicated by a matching star marker. The right subplots indicate that policies from the same neighborhood behave similarly (diagonal segments of the curve) until the failing policy makes a sudden misstep and collects low rewards (horizontal segments).

ALE [7] in Appendix A.5 and A.6. Additional experimental details, including the hyperparameters and implementations used for running these algorithms, can be found in Appendix A.1.

Figure 2 illustrates how different policies produced by deep RL algorithms correspond to a wide range of post-update return distributions, as measured by our chosen metrics [2]. For each metric, we report the bootstrapped mean using 1000 resamples to account for sampling error in the post-update returns collected for a given policy, and omit the corresponding bootstrapped confidence intervals for visual clarity, as they are very small. In particular, this scatter plot shows that different policy parameters achieve similar levels of returns (as measured by the distribution mean) but a wide range of possible levels of variability, as measured by standard deviation, skewness and left-tail probability. This suggests, in a similar way to the example shown in Figure 1, that algorithms discover behaviors which can be qualitatively very different from one another, and that leveraging the post-update return distribution can offer a new lens to investigate different dimensions of policy quality.

These results suggests that simply optimizing the mean return of a policy might ignore its distributional aspect. In particular, a practitioner will likely prefer, for a given level of return, a policy featuring a post-update return distribution with lower levels of standard deviation or left-tail probability. Intuitively, such a policy may correspond to a safer behavior, both able to more robustly accommodate additional updates from its training algorithm and possibly to deal with other unexpected sources of perturbation during deployment.

## 3.2 Analyzing Failures

One characteristic feature of the post-update distributions studied above is the existence of a significant lower tail for many policies visited by the three deep RL algorithms TD3, SAC and PPO. This is visible in their skewness, but especially in their left-tail probability, which demonstrates that many policies produce returns which are unexpectedly poor after up to roughly 10% of updates. We now take a closer look at the specific mechanism by which small changes in an agent's actions results in a wide range of returns in continuous control.

Our experimental procedure is as follows. For each environment, we randomly select 10 policies from the logged checkpoints of 20 independent runs of TD3, conditioned on the fact that the policy has a left-tail probability which is greater than zero. These are policies that we know are prone to poor returns following an update. For each policy, we compute the post-update return distribution by collecting trajectories in the environment after a single update to the original policy. According to this procedure, we identify two trajectories drawn from the neighborhood around the policy: a successful trajectory, characterized by a return within 10% of the mean of the post-update distribution, and a failing trajectory, characterized by a return of less than 50% of the mode of the post-update distribution, as in the left-tail probability.

---

[2]Note that the LTP is not properly defined for a small number of policies achieving negative return, that appear as points on the right border of the scatter plot.

Our goal is to understand the differences between these successful and failing trajectories in order to explain how long-tail returns occur. To this end, Figure 3 depicts two views of the trajectory data. For each environment, the left subplot considers a single pair of successful/failing trajectories corresponding to one of the chosen policies, and plots the reward per timestep earned in these two trajectories. These results suggest that the failing policies which make up the tail of the post-update distribution are capable of collecting similar rewards to the successful policies, yet are prone to missteps which result in episode termination (as in `walker2d`) or transition to a low-reward, quasi-absorbing state (as in `halfcheetah`). Figure 4 shows an example of such a misstep in `walker2d`.

We present a broader view of these observations through the right subplots, per-environment, in Figure 3. Here, we plot each of the trajectory pairs as a parametric curve in time. For both the successful and failing trajectories, we compute the return up to time $t$, $R_{\leq t} = \sum_{i=1}^{t} r(s_i, a_i)$. Then, for each value of $t$, we plot $R_{\leq t}$ for the successful and failing trajectories as a point on the curve, allowing us to visualize the simultaneous evolution of both trajectories.

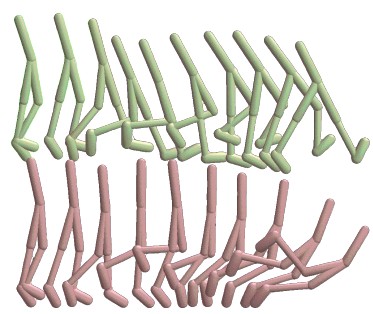

Figure 4: The trajectory of a successful (top) and failing (bottom) policy, both coming from the same post-update distribution in `walker2d`. They exhibit a similar gait until right before the failure.

We assume that $R_{\leq t+1} = R_{\leq t}$ when the length of one trajectory exceeds the other, that is, no additional reward is collected after the trajectory terminates. The resulting visualization reveals several notable findings. First, we show that nearly all trajectory pairs begin by following the line $y = x$, indicating that the respective policies collect rewards at almost exactly the same rate. Next, we observe that many curves rapidly diverge from this line to horizontal, indicating that the failing trajectory suddenly starts collecting little to no reward, while the successful trajectory continues. In `walker2d`, these divergences reflect sudden terminations of the episode, represented by horizontal lines. In `halfcheetah`, which does not terminate, we see that instead the failing agent gets stuck in low-reward absorbing states, but is sometimes able to recover and go back to collecting reward at the same rate as the successful trajectory. We include similar visualizations for the `hopper` and `ant` environments in Appendix A.8, which support the same conclusions.

Taken together, these results demonstrate that some policies exist on the edge of failure, where a slight update can trigger the policy to take actions which push it out of its stable gait and into catastrophe. Indeed, when we compare the gaits learned by policies of high left-tail probability to those which are more well-behaved under updates, we observe that the behaviors of the former are qualitatively more unstable (Figure 1, with more examples in Appendix A.11).

## 4  Navigating Return Landscapes

In the previous section, we took a fine-grained look at the return landscape, using post-update return distributions to characterize the neighborhood of different policies learned by deep RL algorithms. We now consider this landscape on a more global scale, specifically how the agent's return changes as one interpolates between different policies.

### 4.1  Connectivity in the Return Landscape

For our analysis, we use 200 policies generated by different runs of TD3. From these we select pairs of policies with different post-update return distributions, as measured by their standard deviation or left-tail probability, but similar mean. Consider two sets of policy parameters $\boldsymbol{\theta}_1$ and $\boldsymbol{\theta}_2$, for which the post-update return distribution implied by $\boldsymbol{\theta}_1$ has higher LTP than the implied by $\boldsymbol{\theta}_2$. We linearly interpolate these two to form a family of parameters $\boldsymbol{\theta} = \alpha\boldsymbol{\theta}_1 + (1-\alpha)\boldsymbol{\theta}_2, \alpha \in [0, 1]$. For each such $\boldsymbol{\theta}$, we then record the return $R(\boldsymbol{\theta})$ obtained by a single simulation with the corresponding policy.

In Figure 5, we show the result of this interpolation for six pairs of policies in the `hopper` and `walker2d` environments, in two distinct cases. In the first case, the two policies have been produced by the same run of TD3 (i.e., starting from the same initialization and history of batches); in the second case, the two policies have been generated by independent repetitions of the algorithm. The plot shows interesting information about the global structure of the return landscape: the interpolation

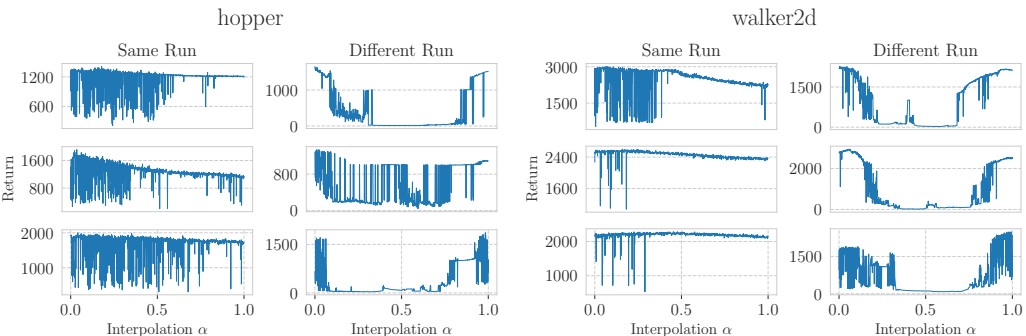

Figure 5: Return of the policies obtained by linear interpolation of the parameters of policies of approximately the same level of return in the `hopper` and `walker2d` environments. The neighborhoods traversed transition from being noisy to being smooth; policies from the same run are connected by paths with no valleys of low performance in the return landscape, even if separated by hundreds of thousands of updates (i.e., at least $1 \times 10^5$ steps for all pairs of policies from the same run).

process traverses different parts of the landscape, highlighting a transition between a noisy part of the landscape to an inherently smoother one. Interestingly, the interpolations between policies from the same run and from different runs exhibit very different qualities. When interpolating between policies of different runs, the process traverses entire regions of the landscape of poor return, until the point in which it gets to the neighborhood of the second policy. By contrast, when interpolating between policies from the same run, the transition from a noisy to a smooth landscape happens without encountering any valley of low return – even when these policies are separated by hundreds of thousands of gradient steps in training. This is particularly surprising given that $\theta$ is a high-dimensional vector containing all of the weights of the neural network, and there is no a priori reason to believe that interpolated parameters should result in policies that are at all sensible.

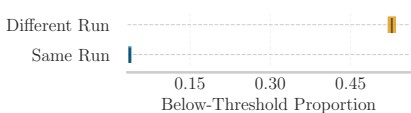

Figure 6: Proportion of return collapses when interpolating between randomly-sampled policies produced by either the same or different runs in Brax. Far fewer return collapses are observed when interpolating between policies produced by the same run. Results are aggregated over all environments with 95% bootstrapped C.I and 500 pairs of policies.

To further quantify the phenomenon, we want to measure the proportion of return collapses encountered when interpolating between policies. We use the following experimental design. We sample for each environment a set of 500 pairs of policies from the same runs and a set of 500 pairs of policies from different runs. Then, we linearly interpolate between policies in the pairs, producing 100 intermediate policies, and randomly perturb them using Gaussian noise with standard deviation $3 \times 10^{-4}$ to obtain an estimate of the mean of their (random) post-update return distribution. Then, for each pair of policies, we measure how frequently the return collapses in between the two extremes, by counting how many times it becomes less than 10% of the minimum return of the two original policies. We then average this *Below-Threshold Proportion* across pairs, and across environments using rliable [1].

Figure 6 shows that there is on average almost no drop in return when interpolating among policies from the same run. We additionally report similar results on four ALE games in Appendix A.6.

We hypothesize this might be interpreted as each individual run of the algorithm specializing on a different family of behaviors, for which, due to the geometry of the return landscape, interpolation between policy parameters does not have any disrupting effect. This result can be interpreted as being related to linear mode connectivity [17, 18], a phenomenon observed in supervised learning, for which different points in the loss landscape of neural networks can be connected by near-constant-loss paths. In other words, it appears there is typically no barrier of low average return separating policies generated from the same run, even when those policies feature very different levels of stability. The existence of such a phenomenon in the RL setting is far from certain: the optimization objective is non-stationary and the evaluation metric (the return instead of the loss) depends on an environment and multiple forward passes from a neural network.

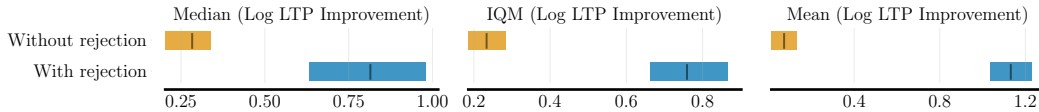

Figure 7: LTP reduction over 40 gradient steps without rejections (TD3) and with rejections (Algorithm 1). Data is aggregated over starting policies, environments, and five independent runs for each starting policy. We see that Algorithm 1 is strictly superior to TD3 with respect to LTP reduction. Results are aggregated over environments with 95% bootstrapped confidence interval.

## 4.2 Stabilizing Policies by Navigating the Landscape

Overall, Figure 5 demonstrates the existence of paths in the return landscape which are able to increase the level of stability of a given policy, but are not necessarily followed in a spontaneous way by typical policy optimization algorithms. In the absence of a desirable end policy to interpolate towards, we would like to understand if it is possible to find similar stabilizing paths (as measured by the LTP), given a starting policy inhabiting a noisy neighborhood of the return landscape. We conjecture that this is feasible by filtering the policy updates produced by an algorithm: In particular, we propose to reject gradient updates that lead to policies with less favorable post-update return distributions.

In our procedure, which is outlined in Algorithm 1, we use the CVaR as a heuristic to compare the stability of post-update return distributions[3], as it is effectively a measure of the left tail mean [38]. Our procedure works as follows: starting with a given policy, we use TD3 to interact with the environment, maintain a replay buffer, and compute updates to the policy and critic parameters. However, before applying a proposed update, we first

**Input:** Initial policy parameter $\boldsymbol{\theta}$, CVaR level $\alpha$, policy update function $\mathfrak{u}$, number of MC samples $N$, tolerance $\delta$
**while** True **do**
$\quad \{R_{\boldsymbol{\theta}}^{(k)}\}_{k=1}^N \overset{\text{iid}}{\sim} \mathcal{R}(\boldsymbol{\theta})$ {Post-update return samples}
$\quad B \leftarrow$ random minibatch
$\quad \boldsymbol{\theta}' \leftarrow \mathfrak{u}(\boldsymbol{\theta}, B)$
$\quad \{R_{\boldsymbol{\theta}'}^{(k)}\}_{k=1}^N \overset{\text{iid}}{\sim} \mathcal{R}(\boldsymbol{\theta}')$
$\quad$ **if** $\text{CVaR}_\alpha(\{R_{\boldsymbol{\theta}'}^{(k)}\}_{k=1}^N) \geq (1-\delta)\text{CVaR}_\alpha(\{R_{\boldsymbol{\theta}}^{(k)}\}_{k=1}^N)$ **then**
$\quad\quad \boldsymbol{\theta} \leftarrow \boldsymbol{\theta}'$
$\quad$ **end if**
**end while**

Algorithm 1: Post-Update-CVaR Rejection

estimate the post-update return distributions of the *post-update* policies by sampling TD3 updates from random minibatches of the replay buffer and evaluating the returns of the corresponding policies. If the estimate of the post-update return CVaR is not sufficiently high relative to that of the post-update return distribution of the current policy, the update is *rejected*, so that the networks and the replay buffer are reverted to the state that they were in before the update was computed. In our experiments, we study the effect that such a rejection mechanism has on the evolution of the LTP by comparing the trajectories induced by this procedure without the ability to reject (i.e., regular TD3) and with the ability to reject.

In Figure 7, we show the improvement in LTP that this algorithm induces when applied to the same policy, aggregated across Brax tasks, using at least 10 policies per environment, after only forty gradient steps. We additionally present scatter plots demonstrating the effect of applying Algorithm 1 to individual policies in Appendix A.10. Our results demonstrate that this rejection procedure can be an effective tool for reducing the LTP of an existing policy.

## 5 Related Work

**Reliability of deep RL.** The goal to avoid catastrophic drops in performance was at the core of the development of foundational methods in deep RL based on conservative updates [41, 42]. Previous work also studied the development of safer algorithms for learning and exploration, both from the theoretical and the empirical standpoints [25, 30, 32, 37, 48]. Our work focuses on understanding the landscape visited by commonly employed policy optimization algorithms and shows that it is

---

[3]See Appendix A.9 for further justification.

possible to relatively easily move from parts of the landscape that induce dangerous behaviors to safer policy parameter vectors. On a higher-level, the sensitivity of deep RL algorithms to stochasticity and hyperparameters, and the extreme variability of results across seeds has been the object of previous studies [2, 11, 20], which mostly focused on proposing more reliable evaluation metrics. Previous work [9] also explicitly advocated for measuring the stability of deep RL algorithms over different axes and using a diverse set of metrics. Our paper proposes a complementary perspective, based on return landscapes and on a distributional view on them. Our procedure which leverages the directions proposed by a policy optimization algorithm to improve the LTP of a policy is related to previous work based on rejection/backtracking strategies [25, 40].

**Return and loss landscapes.** Return landscapes have been previously investigated at a coarser level under the name of reward surfaces/landscapes. In particular, they have been employed for studying the alignment of the gradient directions suggested by policy optimization algorithms to directions of improvement in the actual environment [23] and investigating performance degradation as a long-term optimization danger in such algorithms [43]. Our study of return landscapes with a distributional view in an otherwise fully deterministic setting sheds new light both on the landscape itself and on how it can be leveraged to characterize individual policies. More generally, the investigation of the return that policies collect in an environment is related to the study of the loss landscape of neural networks in supervised learning, for which different techniques have been proposed [28]. Those techniques, together with RL-specific tools, have been employed to explore the loss landscapes of RL algorithms, by visualizing them [5], probing their interaction with entropy regularization [3] or larger neural networks [35]. Our discovery of how policies from the same run are connected by simple paths in parameter space is related to (linear) mode connectivity, which shows a similar behavior in the landscapes of neural networks trained in supervised learning tasks [12–14, 17, 18]. Finally, our work is related to *distributional RL* [6], but we specifically focus on the post-update return distribution as opposed to the distribution of returns under a given policy.

# 6   Discussion and Future Work

In this paper, we have investigated return landscapes in continuous control tasks, as traversed by deep RL algorithms. We demonstrated the existence of noisy neighborhoods of the landscape, where a single update to the policy parameters produces a wide-breadth of post-update returns. By taking a distributional view on these neighborhoods, we revealed the existence of neighborhoods of similar mean return, yet different statistics, which correspond to qualitatively different agent behaviors. We studied the characteristics of failing policies and trajectories in such neighborhoods and attributed their subpar performance to sudden collapses in trajectory reward, rather than overall degradation in the policy. By focusing on linear paths through the global policy landscape, we showed that the landscape exhibits macro-scale variations which extend beyond specific local neighborhoods, and that policies from the same run can be surprisingly connected by linear paths with no valleys of low return. Finally, we demonstrated a simple procedure which discovers paths towards smoother regions of the landscape, starting from a trained policy.

Our results suggest that some of the previously-observed reliability issues in deep reinforcement learning agents for continuous control may be due to the fundamental structure of the return landscape for neural network policies. In particular, while the return of policy in a given neighborhood may be adequate, the distributional structure of the neighborhood characterizes additional dimensions of policy quality: How stable is this policy? What kind of behavior has the agent learned? Is it safe to perform further optimization of this policy? These nuances indicate the potential utility of a landscape-inspired approach to the design of reliable deep RL algorithms.

In addition, our study of parameter interpolation on the return landscape reveals new curiosities surrounding the training behavior of deep reinforcement learning agents. It appears that many policies from the same run fall within a single basin of the return landscape; we conjecture that this may correspond to the algorithm "specializing" on one particular behavior. Our demonstration of regions of lower and higher variability in returns along such paths further supports the possibility of robustifying existing policies, yet also raises the question of whether there are significantly different behaviors separated by barriers of low return, and whether our algorithms can find them. As they are beyond the scope of this paper, we reserve such questions for future work.

## 7 Acknowledgements

The authors thank Jesse Farebrother, Georg Ostrovski, David Meger, Rishabh Agarwal, Josh Greaves, Max Schwarzer and Pablo Castro for insightful discussions and useful suggestions on the early draft, the Mila community for creating a stimulating research environment, and the Digital Research Alliance of Canada for computational resources. This work was partially supported by CIFAR, Fonds de recherche du Québec (FRQNT) and Gruppo Ermenegildo Zegna.

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

# A Appendix

## A.1 Hyperparameters and implementation

For our analyses, we run TD3 and SAC for one million environment steps, with the default hyperparameters reported in Table 1, and collected checkpoints at $5 \times 10^4, 15 \times 10^4, 25 \times 10^4, 35 \times 10^4, 45 \times 10^4, 55 \times 10^4, 65 \times 10^4, 75 \times 10^4, 85 \times 10^4, 95 \times 10^4$ steps for 20 different seeds. We run 20 seeds of PPO for 60 million environment steps with the hyperparameters in Table 1, to obtain a comparable level of performance, being that PPO is notoriously less sample-efficient than the two other algorithms [19]. We use equally-spaced checkpoints, at approximately $6 \times 10^5, 12 \times 10^5, 18 \times 10^5, 24 \times 10^5, 30 \times 10^5, 36 \times 10^5, 42 \times 10^5, 48 \times 10^5, 54 \times 10^5, 60 \times 10^5$ environment steps.

We acknowledge the Python community [34, 46] for developing the core set of tools that enabled this work, including JAX [4, 8], Jupyter [26], Matplotlib [22], numpy [33, 45], pandas [36, 47], and SciPy [24]. Our implementations of TD3 and SAC are based on `jaxrl` [27] and our implementation of PPO is based on CleanRL [21].

Our code is available at `https://github.com/nathanrahn/return-landscapes`.

## A.2 Details of return landscape visualization

To generate the return landscapes in Figure 1, we start from a given policy of parameters $\boldsymbol{\theta}_0$ (produced, in that case, by SAC), and update it two times independently to obtain $\boldsymbol{\theta}'$ and $\boldsymbol{\theta}''$. Then, we plot $R(\boldsymbol{\theta}), \boldsymbol{\theta} = \alpha(\boldsymbol{\theta}' - \boldsymbol{\theta}_0) + \beta(\boldsymbol{\theta}'' - \boldsymbol{\theta}_0) + \boldsymbol{\theta}_0, \alpha \in [-3, 3], \beta \in [-3, 3]$, combining the two gradient directions into other directions to explore the neighborhood. The resulting directions are equivalent to the ones that would be obtained by oversampling or undersampling some elements in a larger batch composed of elements from both the two sampled updates.

## A.3 Interpolation between policies

We now specify the details of the experimental protocol used for Figure 5 and report additional results. In Figure 9, we report results of interpolation for the other two Brax environments, `ant` and `halfcheetah`, missing from Figure 5 for space reasons. We generate the checkpoints for these plots to be policies with similar level of return but different standard deviation of the post-update return distribution. A list of the checkpoints and seeds from TD3 used for producing these plots is shown in Table 2. While the view provided in Figure 5 and Figure 9 provides explicit visual cues on the variability of the neighborhood, it does not leverage directly the post-update return distribution to characterize the neighborhood of the policies obtained by interpolation. To provide a more detailed view, we estimate the post-update return distribution of an unstructured update, by sampling, given a policy of parameters $\boldsymbol{\theta}$, 1000 perturbations from an isotropic Gaussian distribution $\mathcal{N}(\boldsymbol{\theta}, 0.0003)$ and evaluating the resulting policies. Figure 8 shows the mean and the standard deviation (as an error band) of the post-update return distribution of the interpolated policies from the same runs shown in Table 2. This alternate view confirms the results: linearly interpolating between policies of similar mean for the post-update return distribution produces paths of of approximately constant (or smoothly increasing/decreasing) mean post-update return, even when the standard deviation of their post-update return distributions is different.

## A.4 Instabilities in states other than the starting one

In our analyses, we only investigated the return landscape by computing the return on a fixed initial state. Even if the characteristics of the return from other states naturally propagate back to the initial state, it is natural to wonder whether the failures analyzed in our paper also happen naturally in other states. To investigate this question, we compute the LTP of the post-update return distribution for 3 runs of TD3, when starting from one of the states collected by the algorithm during training. In Figure 10, for the `ant` and `walker2d` tasks, we show for each of these states, identified by their collection time (index in the replay buffer) the LTP averaged over all the checkpoints collected during training. The results indicate that, regardless of some variation across seeds, there appears not to be any special pattern in the presence of failures over states grouped by time, and that the tail of the post-update distributions is long even when measured from states other the initial one.

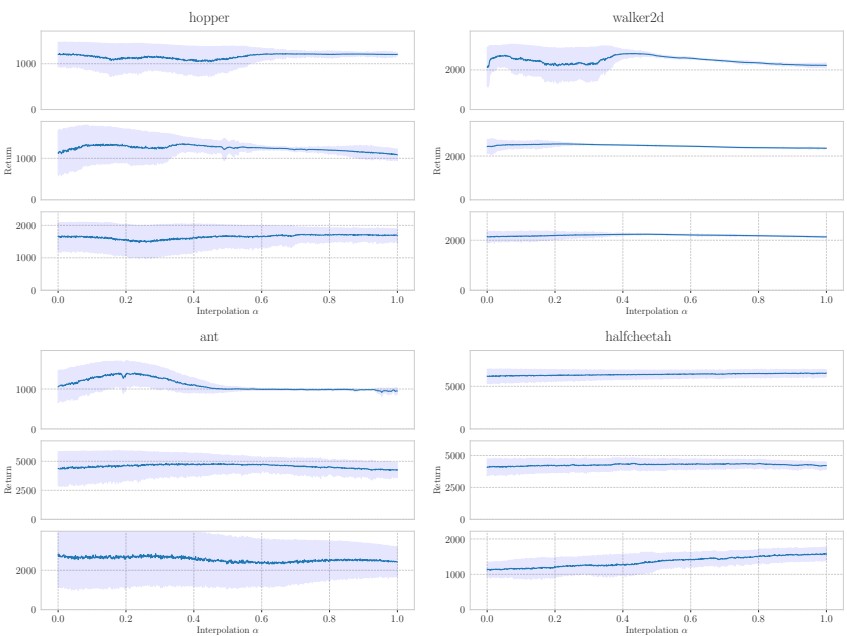

Figure 8: Mean and standard deviation, displayed as error band, of the (noise-based) post-update return distributions for the four different Brax environments. The mean of the post-update return distribution remains approximately constant when linearly interpolating between checkpoints of similar level of return.

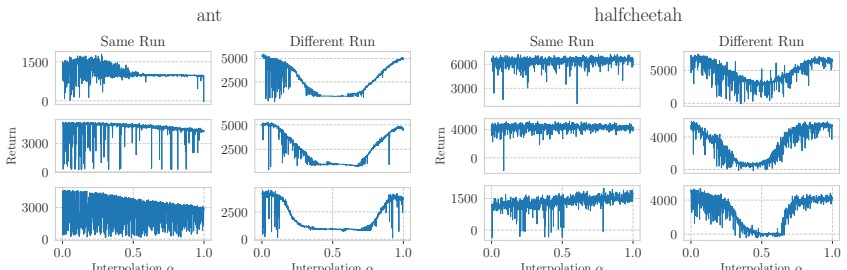

Figure 9: Results on linear interpolation among policies in the `ant` and `halfcheetah`.

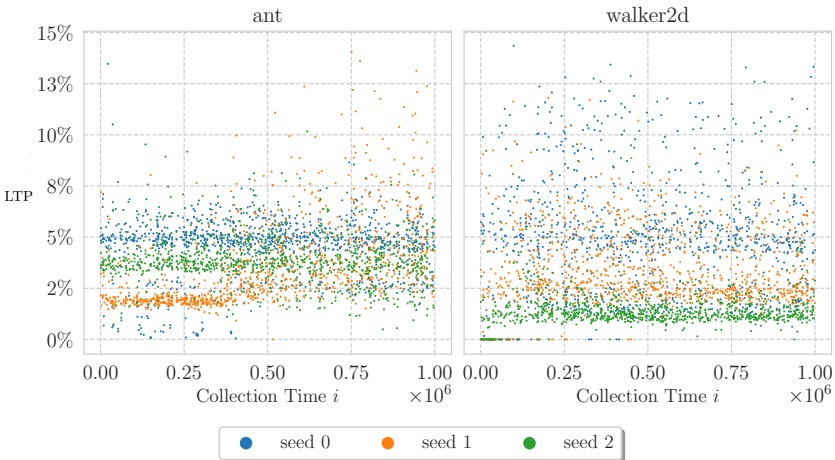

Figure 10: Left-tail probability computed on different states, from a replay buffer for 3 seeds of TD3, averaged across all checkpoints.

Table 1: Hyperparameters for TD3, SAC and PPO.

| TD3 | |
|---|---|
| **Parameter** | Setting |
| Discount factor | 0.99 |
| Minibatch size | 256 |
| Optimizer (all) | Adam |
| Optimizer (all): learning rate | 0.0003 |
| Optimizer (all): $\beta_1$ | 0.9 |
| Optimizer (all): $\beta_2$ | 0.999 |
| Optimizer (all): $\epsilon$ | 0.00015 |
| Networks (all): activation | ReLU |
| Networks (all): n. hidden layers | 2 |
| Networks (all): hidden units | 256 |
| Replay Buffer Size | $10^6$ |
| Updates per step | 1 |
| Target update period | 1 |
| $\tau$ (EMA coefficient) | 0.995 |
| Exploration noise | 0.1 |
| Actor Delay | 2 |
| Policy Noise | 0.2 |
| Noise Clip | 0.5 |

| SAC | |
|---|---|
| **Parameter** | Setting |
| Discount factor | 0.99 |
| Minibatch size | 256 |
| Optimizer (all) | Adam |
| Optimizer (all): learning rate | 0.0003 |
| Optimizer (all): $\beta_1$ | 0.9 |
| Optimizer (all): $\beta_2$ | 0.999 |
| Optimizer (all): $\epsilon$ | 0.00015 |
| Networks (all): activation | ReLU |
| Networks (all): n. hidden layers | 2 |
| Networks (all): hidden units | 256 |
| Initial Temperature | 1 |
| Replay Buffer Size | $10^6$ |
| Updates per step | 1 |
| Target update period | 1 |
| $\tau$ (EMA coefficient) | 0.995 |

| PPO | |
|---|---|
| **Parameter** | Setting |
| Discount factor | 0.99 |
| Optimizer (all) | Adam |
| Optimizer (all): learning rate | 0.0026 |
| Optimizer (all): $\beta_1$ | 0.9 |
| Optimizer (all): $\beta_2$ | 0.999 |
| Optimizer (all): $\epsilon$ | 0.00001 |
| Networks (all): activation | ReLU |
| Networks (all): n. hidden layers | 2 |
| Networks (all): hidden units | 256 |
| Parallel Envs | 2048 |
| Steps to update | 16 |
| GAE $\lambda$ | 0.95 |
| Num. Updates | 4 |
| Max Grad. Norm | 1 |

### A.5 Scatter plots of statistics of post-update return distribution

We now show similar "maps" of the landscape to Figure 2 for all of the Brax environments we study. Results are shown in Figure 11, Figure 12, and Figure 13. We also include similar results for the `quadruped-walk`, `quadruped-run`, `humanoid-stand`, `humanoid-walk`, and `humanoid-run` tasks from DeepMind Control Suite [44] in Figures 14, 15, 16, 17, and 18. We believe that this type of scatter plot can provide a different perspective, compared to the one only focused on training curves, to study the interaction between a particular algorithm and an environment.

### A.6 Discrete control

Despite the fact that the main focus of our paper is on continuous control, we also run small-scale experiments to understand if the tools we introduced generalize to discrete-action environments. In particular, we leverage four games from the standard ALE benchmark: `Qbert`, `MsPacman`, `Breakout`, and `Seaquest` [7]. We run PPO on these environments for 5 runs and for 10 million steps each, and measure post-update return distribution statistics using the same protocol employed in the rest of

Table 2: List of checkpoints used for Figure 5, Figure 9 and Figure 8.

| **Same Run** | | | **Diff. Run** | | | |
|---|---|---|---|---|---|---|
| Seed | Ckpt 1 | Ckpt 2 | Seed 1 | Ckpt 1 | Seed 2 | Ckpt 2 |
| `ant` | | | `ant` | | | |
| 1 | 550000 | 150000 | 14 | 850000 | 4 | 750000 |
| 6 | 850000 | 650000 | 11 | 750000 | 4 | 550000 |
| 11 | 650000 | 250000 | 2 | 650000 | 4 | 350000 |
| `halfcheetah` | | | `halfcheetah` | | | |
| 3 | 250000 | 950000 | 10 | 950000 | 7 | 450000 |
| 6 | 250000 | 350000 | 7 | 750000 | 16 | 750000 |
| 13 | 250000 | 550000 | 1 | 250000 | 6 | 350000 |
| `hopper` | | | `hopper` | | | |
| 2 | 450000 | 250000 | 7 | 350000 | 4 | 450000 |
| 15 | 550000 | 150000 | 1 | 250000 | 2 | 150000 |
| 17 | 950000 | 750000 | 9 | 550000 | 6 | 250000 |
| `walker2d` | | | `walker2d` | | | |
| 1 | 750000 | 250000 | 14 | 350000 | 17 | 450000 |
| 4 | 950000 | 850000 | 1 | 550000 | 10 | 950000 |
| 4 | 550000 | 450000 | 17 | 350000 | 3 | 250000 |

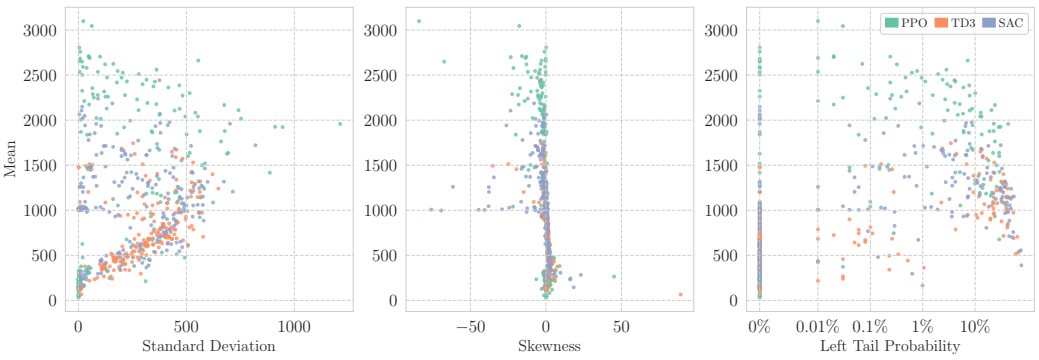

Figure 11: A scatter plot showing mean return and standard deviation, skewness or left-tail probability of the post-update return distribution of policies produced by three popular deep RL algorithms on the `halfcheetah` Brax task.

Figure 12: A scatter plot showing mean return and standard deviation, skewness or left-tail probability of the post-update return distribution of policies produced by three popular deep RL algorithms on the `hopper` Brax task.

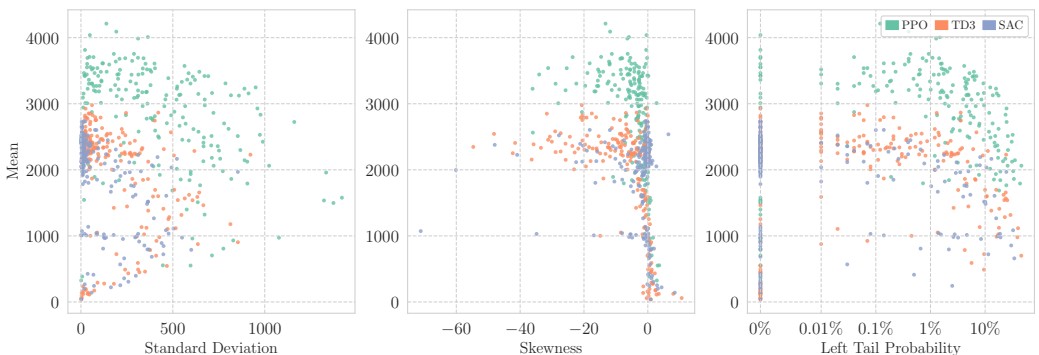

Figure 13: A scatter plot showing mean return and standard deviation, skewness or left-tail probability of the post-update return distribution of policies produced by three popular deep RL algorithms on the `walker2d` Brax task.

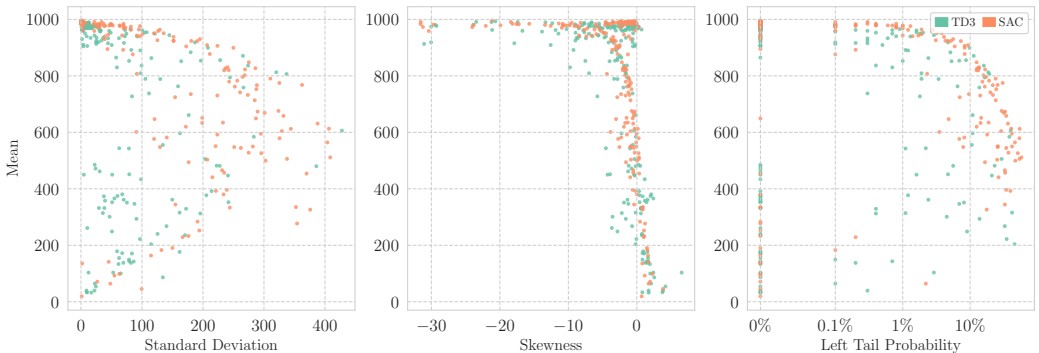

Figure 14: A scatter plot showing mean return and standard deviation, skewness or left-tail probability of the post-update return distribution of policies produced by two popular deep RL algorithms on the `quadruped-walk` DeepMind Control task.

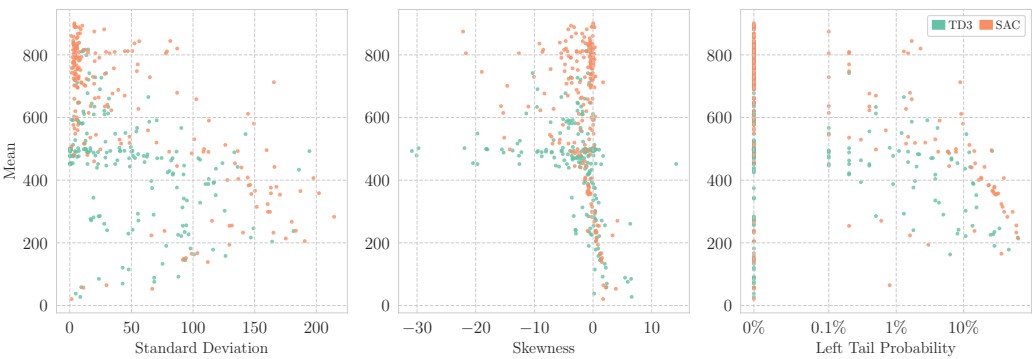

Figure 15: A scatter plot showing mean return and standard deviation, skewness or left-tail probability of the post-update return distribution of policies produced by two popular deep RL algorithms on the `quadruped-run` DeepMind Control task.

our paper: we evaluate 10 policies evenly-spaced across training, and perform 1000 independent updates to each policy. We find that post-update return distributions computed for these environments still exhibit a remarkable degree of variation, as captured by their standard deviation (see Figure 21). At the same time, the shape of the resulting distributions can be quite different compared to the ones observed in robotic locomotion tasks, and, thus, it is not necessarily described in a rich way by metrics such as the LTP (see Figure 20).

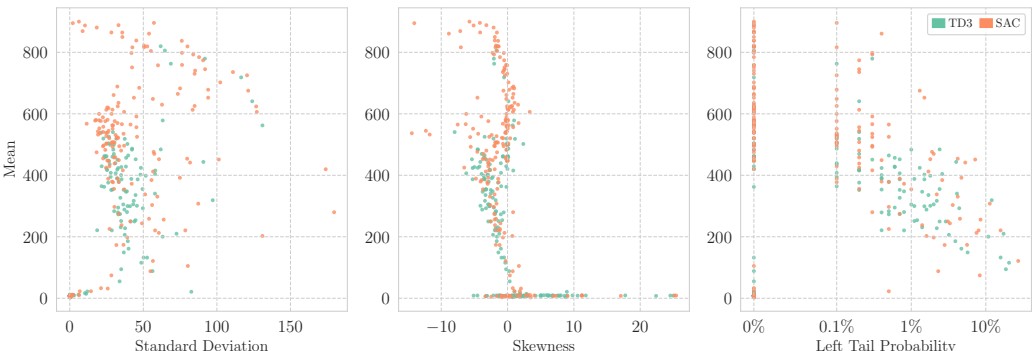

Figure 16: A scatter plot showing mean return and standard deviation, skewness or left-tail probability of the post-update return distribution of policies produced by two popular deep RL algorithms on the `humanoid-stand` DeepMind Control task.

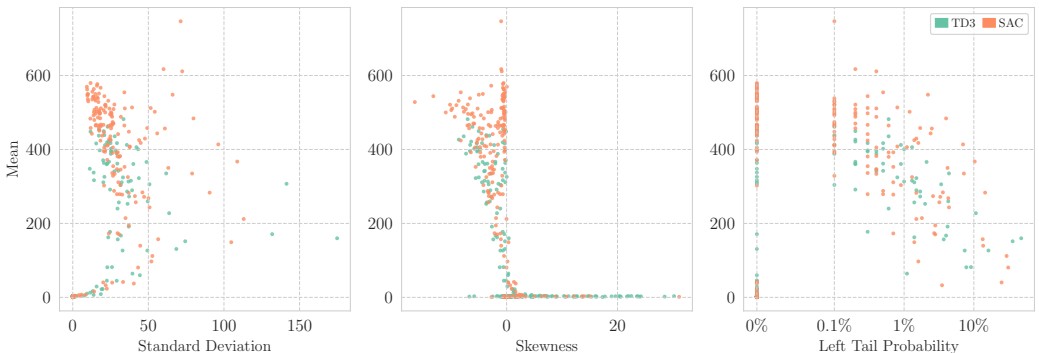

Figure 17: A scatter plot showing mean return and standard deviation, skewness or left-tail probability of the post-update return distribution of policies produced by two popular deep RL algorithms on the `humanoid-walk` DeepMind Control task.

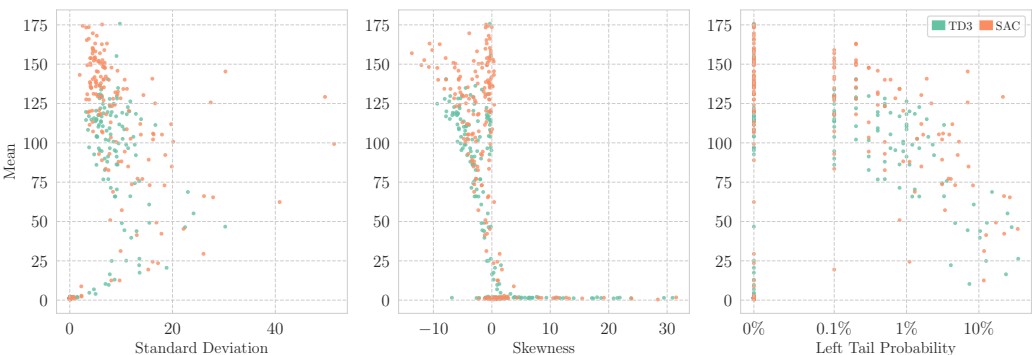

Figure 18: A scatter plot showing mean return and standard deviation, skewness or left-tail probability of the post-update return distribution of policies produced by two popular deep RL algorithms on the `humanoid-run` DeepMind Control task.

We also run an interpolation experiment on the ALE to investigate whether the connectivity phenomenon studied in Section 4.1 generalizes to discrete-control environments. We produce quantitative results on the phenomenon with the following procedure. For each one of the four games, we sample a set of at least 20 pairs of policies from same runs and 20 pairs of policies from different runs of the algorithm. Then, we linearly interpolate between policies in the pairs, producing 50 intermediate policies, and randomly perturb them using Gaussian noise with standard deviation 0.0003 to obtain an estimate of the mean of their (random) post-update return distribution. Then, for each pair of policy, we measure how frequently the return collapses in between the two extremes, by counting

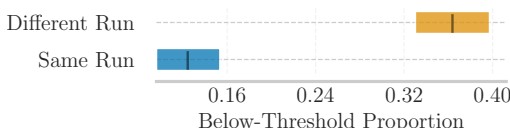

Figure 19: Proportion of return collapses when interpolating between randomly-sampled policies produced by either the same or different runs in the ALE (`QBert`, `Breakout`, `Ms.Pacman`, `Seaquest`). Far fewer return collapses are observed when interpolating between policies produced by the same run. Results are aggregated over environments with 95% bootstrapped C.I..

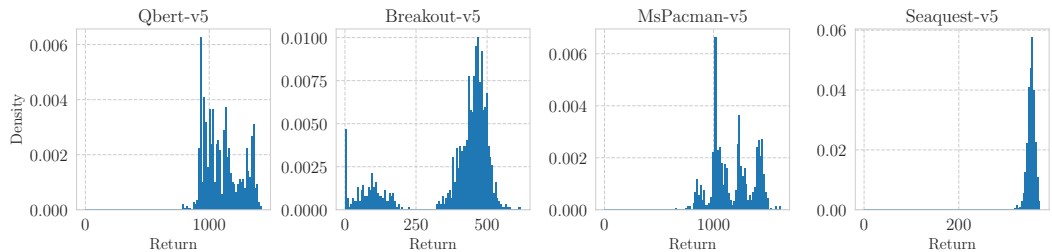

Figure 20: Histograms of the post-update return distribution for policies obtained by PPO on 4 Atari environments. These distributions were selected for illustrative purposes to demonstrate that 1) returns can vary significantly after a single policy update (see also Figure 21) and 2) the distributions obtained can exhibit varied profiles.

how many times it becomes less than 10% of the minimum return of the two original policies. We then average this Below-Threshold Proportion across pairs, and across environments using rliable [1]. Figure 19 shows that the phenomenon, properly quantified, is still present when using a very different class of environments (discrete-action, game-based).

### A.7 Behavior cloning

While the results in the main paper demonstrate that policies learned by commonly-used RL algorithms are sensitive to updates, we were curious to understand whether behavior cloning produces policies which occupy less noisy neighborhoods of the return landscape. To do so, we conducted a set of additional experiments on 4 Brax environments. The protocol was as follows: For each environment, we consider 10 independent training runs of TD3, and 5 policies distributed evenly throughout the run. For each of these policies, we train a new agent using behavior cloning on the data logged up until the collection time of the teacher policy, replacing the actions in the dataset with the actions of the teacher policy, for 1 million gradient steps. We log 10 policies throughout each training run of behavior cloning. To compute the post-update return distribution for the policies obtained by behavior cloning, we used one additional gradient step on the MSE-based BC objective, and 1000 samples.

In Figure 22 (left), we compare statistics of the post-update return distributions for all pairs of policies $(\pi_{i,j}^{BC}, \pi_i^{TD3})$ where policy $\pi_{i,j}^{BC}$ is obtained by behavior cloning $\pi_i^{TD3}$. We compute the Pearson correlation coefficient of statistics of the post-update return distributions of these policies: between the mean of each pair, and between the LTPs of each pair. We find that the means are highly correlated and that the learned BC policies are comparable in performance to their teacher policy. We additionally show that correlation in the LTP is much more variable across environments – in general, cloning a policy of high or low LTP does not always lead to a cloned policy of the same LTP.

But does training policies by BC produce policies occupying less noisy neighborhoods of the return landscape, which would have correspondingly lower LTP overall? In Figure 22 (right), we show the mean LTP across policies and environments, along with its 95% bootstrapped confidence interval following the recommendations from *rliable* [1]. Our results demonstrate that BC does not produce fundamentally more stable policies, as measured by the LTP.

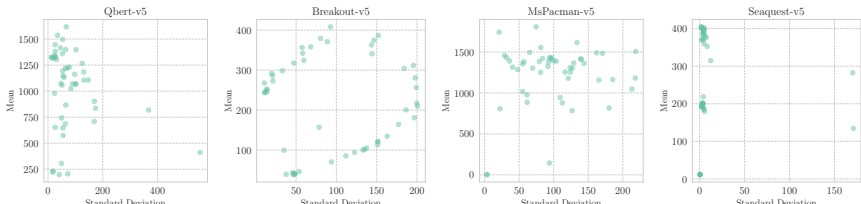

Figure 21: Scatter plots of the mean and standard deviation of the post-update return distribution of policies learned by PPO across four Atari environments. In the majority of cases, the post-update return distribution exhibits significant variability relative to its mean, indicating that learned policies in Atari are also sensitive to single updates.

|             | $corr_{mean}$ | $corr_{LTP}$ |
|-------------|---------------|--------------|
| ant         | 0.994         | 0.763        |
| halfcheetah | 0.995         | 0.976        |
| walker2d    | 0.973         | 0.218        |
| hopper      | 0.753         | 0.449        |

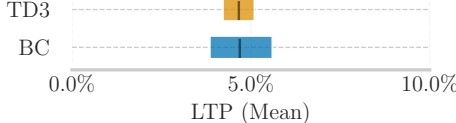

Figure 22: Comparing pairs of policies obtained by behavior cloning a given TD3 policy. **Left:** BC learns policies which are well-correlated in mean return with the cloned policy, measured by Pearson's correlation coefficient. The correlation in LTP is more variable across environments, indicating that cloning a given policy does not always produce a new policy of the same LTP. **Right:** Average LTP of policies learned by TD3 and BC, aggregated across environments. Overall, BC learns policies of similar LTP to policies learned by TD3. Results are aggregated over environments with 95% bootstrapped confidence interval.

## A.8 Complete results on comparing successful and failing trajectories

In Figure 23, we include similar results to Figure 3 for all the Brax environments we study.

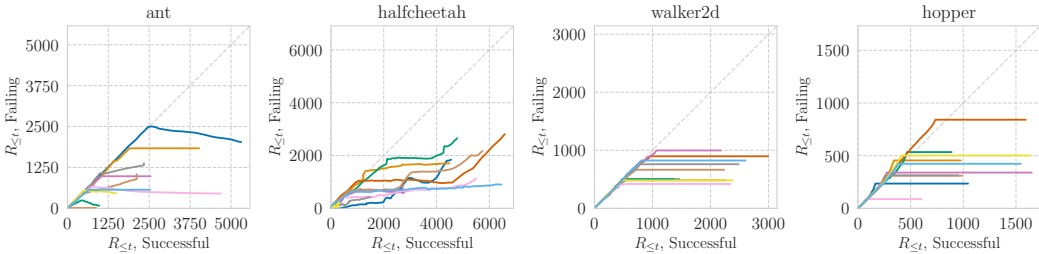

Figure 23: A visualization of how failures occur on four Brax tasks. The plots show the simultaneous evolution of returns for 10 trajectory pairs, of one successful and one failing drawn from the same post-update policy/trajectory distribution. We observe that these policies from the same neighborhood behave similarly (diagonal segments of the curve) until the failing policy makes a sudden misstep and collects low rewards (horizontal segments). At divergence, we observe that the failing policy can terminate, collect no reward, or even collect negative reward, yet it sometimes recovers and goes back to matching the performance of the successful policy.

## A.9 Justification of rejection procedure

In Algorithm 1, we proposed a procedure that rejects gradient updates based on estimates of the CVaR of the resulting policies' post-update returns. Since our goal with this procedure is ultimately to decrease the LTP of an existing policy, one might rightfully question why CVaR was our heuristic of choice. Our reasoning for this choice was simply that the LTP can be minimized by arbirarily poor policies: that is, policies that only achieve "low enough" returns will have very low LTP, because they cannot substantially fail relative to their existing performance. Such an issue can be circumvented by comparing the CVaR of the post-update returns: among all policies that have the same left-tail

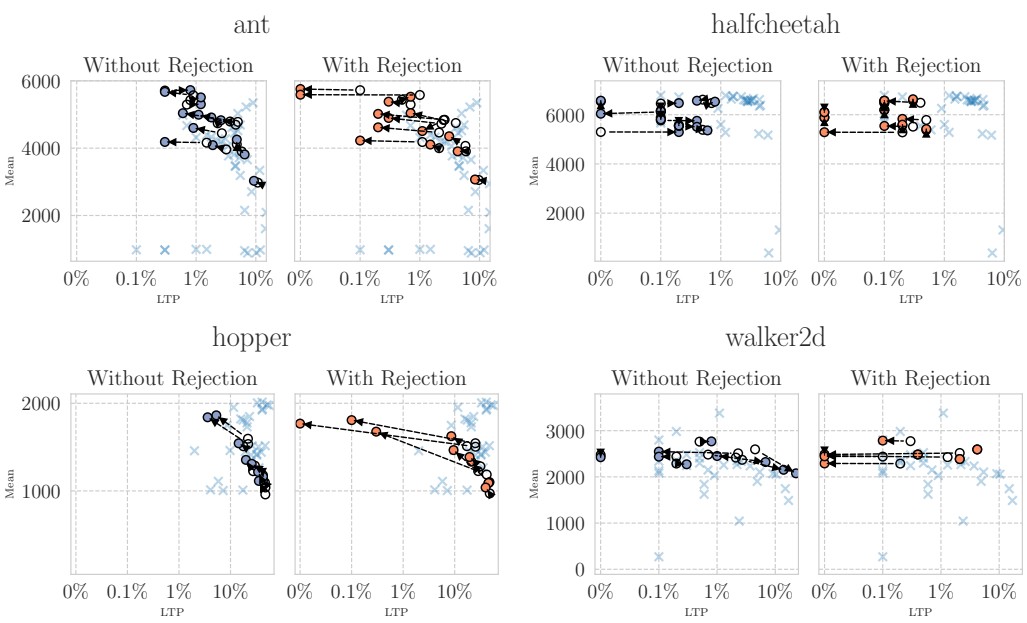

Figure 24: Experimental results for Algorithm 1 on 4 Brax domains. Each empty circle denotes the starting policy, while the filled circle at the end of the arrow shows the policy obtained after application of Algorithm 1. X's show other policies obtained by TD3, for context.

probability, the CVaR is still able to order the policies by the quality of the returns *in* the left tail. To further justify the use of CVaR as a rejection heuristic, we visualized how different CVaR heuristics order policies with respect to their post-update return distributions. Figure 25 illustrates how the $\alpha$-CVaR ranks policies. Note that $\alpha$-CVaR in this instance is effectively measuring the mean among the post-update returns below the $\alpha$-quantile of the post-update return distribution, so 0-CVaR would measure the worst-case post-update return and 1.0-CVaR is equivalent to the mean of the post-update returns.

As expected, we see that ranking by the mean is relatively agnostic to the LTP of the policies, but for $\alpha < 1$, ranking by CVaR is effectively trading off larger mean post-update returns for lower LTP. Notably, the gradient of colors (rankings) in the 0.01-CVaR column is distinctly "top-left to bottom-right", indicating that only policies with both high returns and low LTP are ranked favorably. The 0.1-CVaR column is fairly similar, but the gradient is more vertical – that is, it is more forgiving with respect to LTP than the 0.01-CVaR heuristic, but it still clearly prefers policies with lower LTP among those with similar levels of return. Finally, the gradient in the rightmost column is exactly vertical, meaning that the ranking is agnostic to LTP.

Consequently, we argue that CVaR is a reasonable heuristic for identifying robust policies without neglecting their performance level. In our experiments, we found that the 0.1-CVaR heuristic for rejecting gradient updates struck the best balance between robustness and performance.

### A.10 Additional post-update-CVaR rejection results

In Figure 24 we include the results from our experiments with Algorithm 1 in four Brax domains for policies found by TD3.

### A.11 Visualization of behaviors

We uploaded some visualization of the behaviors on a website (`https://sites.google.com/view/on-return-landscapes/home`): policies of similar mean for the post-update return but different standard deviation exhibit different behaviors.

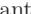

Figure 25: Rankings of policies by the $\alpha$-CVaR of their post-update returns. Points with darker colors are ranked higher, and stars denote the highest ranked policy.

## A.12 Evidence of multi-scale structure

To investigate whether the shape of a neighborhood heavily depends on the size of the steps used for exploring it, we "zoom-in" on the landscape plot from Figure 1 by using the same granularity but on a smaller window. We apply this process twice, with the results in Figure 27 giving evidence for scale-independence.

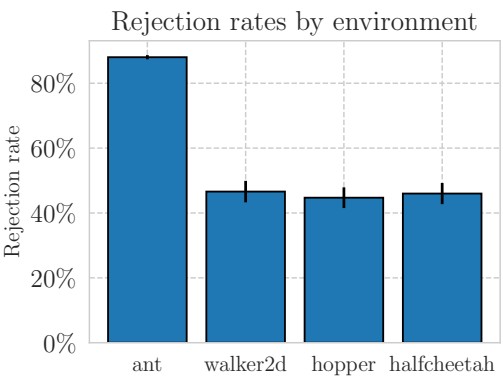

Figure 26: Frequency of rejections during execution of Algorithm 1 across environments. Bars depict the mean across starting policies and runs, and the error bars represent standard error.

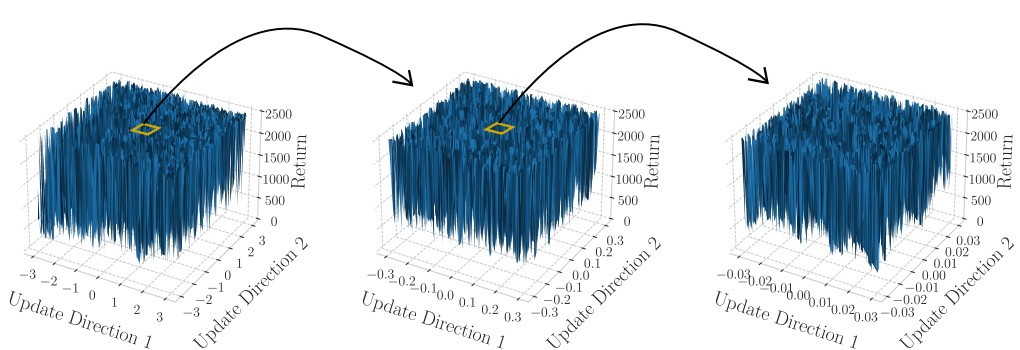

Figure 27: Multiscale version of the plot in Figure 1. Note the similarity between the original and the more granular versions, giving evidence of the independence of the general shape of a neighborhood from the resolution at which it is visualized.

