# OpenReview forum: "Policy Optimization in a Noisy Neighborhood: On Return Landscapes in Continuous Control"
_NeurIPS.cc/2023/Conference — NeurIPS 2023 poster_

### Official Review · Reviewer_Lizq · 2023-07-03

**Soundness:** 2 fair
**Presentation:** 3 good
**Contribution:** 2 fair
**Rating:** 6
**Confidence:** 3

**Summary:**

This paper explores the concept of return landscapes in deep reinforcement learning agents for continuous control. The authors demonstrate that the mapping between policy parameters and return can exhibit high-frequency discontinuities, leading to variations in agent performance. They introduce the concept of noisy neighborhoods in the return landscape and show that different policies can have substantially different distributional profiles, even when they achieve similar average returns. The authors also investigate the impact of noisy neighborhoods on learning dynamics and propose a distribution-aware procedure to improve policy robustness.

**Strengths:**

- The paper introduces a fresh perspective on the instability of deep reinforcement learning agents in continuous control by studying the return landscape.
- The concept of noisy neighborhoods and the characterization of post-update return distributions provide valuable insights into the relationship between policy parameters and return.
- The experiments and visualizations presented in the paper help illustrate the findings and support the claims made by the authors.

**Weaknesses:**

This paper is well-written, and the authors thoroughly explain the key concepts and conclusions by employing extensive experiments and visualizations. However, I believe there are areas for improvement:

1. In Section 4.2, the authors state, "while Figure 5 demonstrates that we can smoothly interpolate between levels of stability along a single gradient direction, Figure 1 illustrates that many gradient directions are not so forgiving. As such, we posit that the stability of policies over the course of training can be improved simply by rejecting gradient updates that lead to policies with less favorable post-update return distributions." I find the introduction of Algorithm 1 to be somewhat abrupt, as it does not seem closely related to the conclusions drawn earlier in the paper. For instance, an important observation made is that "we can smoothly interpolate between levels of stability along a single gradient direction." In algorithm design, can we not only reject policy updates that create noisier neighborhoods but also leverage policy updates that result in more stable neighborhoods?

2. The paper designs a simple yet effective algorithm based on crucial conclusions derived from in-depth analysis. However, due to the algorithm's conciseness and clarity, it becomes difficult for readers to link it to other existing methodologies and place it appropriately within the field. Therefore, a more thorough comparison or discussion with related work is necessary. For example, in the related work section, the authors have mentioned that their work is related to studies based on rejection/backtracking strategies.

**Questions:**

1. I am particularly interested in whether distributional RL algorithms exhibit the issues mentioned in the paper. Given that distributional RL methods take into account reward distribution during policy optimization (albeit with significant differences from the post-update reward distribution studied in this paper), can these methods produce smoother reward landscapes?

2. Can the authors analyze the limitations of Algorithm 1? In particular, are the main conclusions and Algorithm 1 still valid for policies updated using different methods (beyond PPO, SAC, and TD3)?

3. On average, how many times will Algorithm 1 reject an update during a single policy parameter update in actual operation? Will the efficiency of the algorithm be severely affected if the number of rejected updates becomes too high?

**Limitations:**

Limitations are not explicitly discussed by the paper. Some comments and questions regarding limitations of the work are discussed in the section above.

---

> ### Author Rebuttal · Authors · 2023-08-10
>
> Thank you for your feedback!
>
>
> > "while Figure 5 demonstrates that we can smoothly interpolate between levels of stability along a single gradient direction"
>
> We apologize for the confusion. This is a typo in the original version of our paper: the direction that is used to interpolate between policies is not a gradient direction, but it is instead the direction going from one vector to the other in the parameter space.
>
> > can we not only reject policy updates that create noisier neighborhoods but also leverage policy updates that result in more stable neighborhoods?
>
> We appreciate the suggestion. We believe this is indeed an exciting avenue for future research, but we found this approach to be much less stable than the approach presented in Algorithm 1 in our experimentation. Moreover, we leverage Algorithm 1 to show that many, but not all, of the directions proposed by policy optimization algorithms during training are actually sensible directions of improvement in terms of LTP. We believe this observation could be useful for practitioners who wish to improve this class of algorithms.
>
> > Therefore, a more thorough comparison or discussion with related work is necessary. For example, in the related work section, the authors have mentioned that their work is related to studies based on rejection/backtracking strategies
>
> We appreciate the suggestion. We will expand our discussion in the related work section.
>
> > I am particularly interested in whether distributional RL algorithms exhibit the issues mentioned in the paper. Given that distributional RL methods take into account reward distribution during policy optimization (albeit with significant differences from the post-update reward distribution studied in this paper), can these methods produce smoother reward landscapes?
>
> This is an excellent question, and indeed, it was an approach that we experimented with extensively. We experimented with a distributional extension of TD3, where the critic was replaced with a distributional critic. Additionally, we experimented with risk-averse policy updates using the distributional critic, where the Conditional Value at Risk (CVaR) of the return distribution was reinforced rather than the mean return, with the hypothesis that the risk-averse policies should navigate towards smoother neighborhoods. Unfortunately, our findings showed no significant improvement with regard to the smoothness of the neighborhoods encountered by the distributional agents.
>
> We also compared the post-update return distributions of the policies visited by TD3 and distributional TD3, and observed that there is no substantial difference in neighborhoods reached by the distributional algorithms relative to TD3. We found that the distributional critic does not provide a reliable estimate of the post-update return distribution. Indeed, the distributional Bellman equation characterizing the return distributions in distributional RL does not model such an object explicitly.
>
> > Can the authors analyze the limitations of Algorithm 1? In particular, are the main conclusions and Algorithm 1 still valid for policies updated using different methods (beyond PPO, SAC, and TD3)?
>
> To understand the limitations of Algorithm 1 in an extreme case, we have run the same rejection procedure but using random directions proposed by simple Gaussian perturbations to the policy parameters instead of the directions produced by policy optimization algorithms. In this extreme case, we found that Algorithm 1 is no longer an effective procedure for improving the LTP, suggesting that the Algorithm might be sensitive to the presence of bad update directions.
>
> Another limitation of the algorithm, which does not create particular problems for the specific evaluation setting leveraged in our paper but might create problems when aiming at sample efficiency, is the need to obtain rollouts from the environment to evaluate the post-update return distribution statistics and reject an update.
>
> > On average, how many times will Algorithm 1 reject an update during a single policy parameter update in actual operation? Will the efficiency of the algorithm be severely affected if the number of rejected updates becomes too high?
>
> Thanks for the question. We compute the statistics for the frequency of rejections across five seeds in Figure 7 in the pdf.
>
> Indeed, the rate of improvement of the algorithm (with respect to the policy return) may be reduced when using Algorithm 1. However, Algorithm 1 is only meant as a procedure to improve an existing acceptable policy with respect to its LTP -- we are not looking for fast improvement at this stage. Having said that, as shown in Figure 6, the difference in mean return after 40 gradient steps of Algorithm 1 tends to be comparable to that of TD3.

---

> > ### Comment · Reviewer_Lizq · 2023-08-15
> >
> > Thank you for your detailed rebuttal, which has addressed most of my concerns regarding your manuscript. I appreciate the effort and time spent on explaining the key aspects of your work.
> >
> > Regarding the comparison and analysis of the proposed algorithm with existing algorithms, it would be beneficial for you to include a brief yet focused discussion in the rebuttal. If you can clearly highlight the differences and advantages of your algorithm compared to others in the field, it could potentially lead to a higher evaluation score.
> >
> > Incorporating this information in your rebuttal would not only strengthen the manuscript by providing better context for readers but will also aid in emphasizing the novelty and significance of your work. I look forward to your response on this particular aspect.

---

> > > ### Author Response · Authors · 2023-08-17
> > >
> > > Thank you for your response and for providing us the opportunity to clarify the relationship between ours and related work. We agree that adding further discussion about the distinctions between Algorithm 1 and existing algorithms in the literature will strengthen the paper.
> > >
> > > Our Algorithm is best contextualized against existing methods based on 1) policy gradient methods which optimize a risk criterion and 2) rejection sampling.
> > >
> > > Methods in the first class mainly aim to improve the stability of a policy by optimizing against a distributional critic [7], such as WCSAC or SDAC [2, 6]. Importantly, these algorithms do not consider the return landscape: They estimate only the distribution of returns for a single policy parameter and therefore do not produce policies which are robust under updates. In our experiments, we found that estimation of the true post-update return distribution with distributional RL techniques is quite difficult, and that the resulting algorithms did not reliably produce policies of improved LTP. Therefore, an advantage of Algorithm 1 is that it does not depend on any risk-sensitive algorithm design specified *a priori*, since these approaches typically require a well-behaved distributional critic and risk-sensitive objective to optimize. In contrast, we applied Algorithm 1 on top of TD3 policies that were trained with standard hyperparameters.
> > >
> > > Our algorithm is most closely related to works based on rejection sampling. In the context of reinforcement learning, previous work made use of rejection sampling in the action space, with respect to a critic, in order to choose safe actions [3, 4]. Instead, our Algorithm 1 performs rejection in the space of policy parameters, related to success-story algorithms [5], in which updates to the policy are discarded if they lead to worse average returns. A key related work based on selective improvements to policies is the EVEREST algorithm of [1]. Our Algorithm 1 admits several advantages when compared to their procedure: First, EVEREST is based on the rejection of policies which do not probably improve the return, but it does not produce policies which are inherently stable under further updates by considering the return landscape around a parameter vector. Second, EVEREST performs rejection at a frequency on the order of once per thousands of updates; our algorithm is instead able to significantly improve the LTP over a small number of update steps (40).
> > >
> > >
> > > [1] Khanna, Pranav, et al. "Never Worse, Mostly Better: Stable Policy Improvement in Deep Reinforcement Learning." arXiv preprint arXiv:1910.01062 (2019).
> > >
> > > [2] Yang, Qisong, et al. "WCSAC: Worst-case soft actor critic for safety-constrained reinforcement learning." Proceedings of the AAAI Conference on Artificial Intelligence. Vol. 35. No. 12. 2021.
> > >
> > > [3] Bharadhwaj, Homanga, et al. "Conservative Safety Critics for Exploration." International Conference on Learning Representations. 2021.
> > >
> > > [4] Srinivasan, Krishnan, et al. "Learning to be safe: Deep RL with a safety critic." arXiv preprint arXiv:2010.14603 (2020).
> > >
> > > [5] Schmidhuber, Jürgen, Jieyu Zhao, and Marco Wiering. "Shifting inductive bias with success-story algorithm, adaptive Levin search, and incremental self-improvement." Machine Learning 28 (1997): 105-130.
> > >
> > > [6] Kim, Dohyeong, Kyungjae Lee, and Songhwai Oh. "Efficient Trust Region-Based Safe Reinforcement Learning with Low-Bias Distributional Actor-Critic." arXiv preprint arXiv:2301.10923 (2023).
> > >
> > > [7] Bellemare, Marc G., Will Dabney, and Rémi Munos. "A distributional perspective on reinforcement learning." International conference on machine learning. PMLR, 2017.

---

> > > > ### Comment · Reviewer_Lizq · 2023-08-19
> > > >
> > > > Thank you for your detailed response. I have increased the score from 5 to 6.

---

### Official Review · Reviewer_Voa2 · 2023-07-05

**Soundness:** 2 fair
**Presentation:** 4 excellent
**Contribution:** 3 good
**Rating:** 7
**Confidence:** 3

**Summary:**

The paper investigates the mapping from policy parameters $\theta$ to returns $R(\theta)$ and the sensitivity of $R(\theta)$ to minor perturbations to $\theta$. The authors show that popular RL algorithms tend to traverse "noisy neighborhoods" in policy parameter space where the return is highly varied even for small changes to $\theta$. They find that policies with comparable average returns can have drastically different return distributions. Additionally, interpolating between policies from the same training run allows for transitioning from noisy to smooth return landscapes without degraded performance (the same is not true for policies from different runs). Finally, a practical algorithm is proposed which improves stability by rejecting gradient updates that lead to noisier return distributions.

**Strengths:**

Overall, I thought the paper was enjoyable to read and appears fairly novel (to the best of my knowledge). The figures are great and in general the story of the paper flows very well; Figure 2 is fantastic.

The landscape-inspired approach to deep RL is extremely interesting in my opinion. Finding novel ways to stabilize training by considering the local return landscape around the current set of policy parameters is highly relevant to scaling up and robustifying existing RL methods.

I appreciate that the paper doesn't rely on overly technical jargon and, combined with the great illustrations, the paper is fairly easy to grasp without much math at all.

**Weaknesses:**

Section 4.1 makes pretty strong claims about interpolating between policies from the same run and from different runs, but it is only really substantiated by a few (six) qualitative examples. I suspect there is some way to quantify the claims in this section across a larger population of policy pairs, which I think would significantly strengthen this section.

The results in Figure 6 (and the description in Section 4.2) are a bit confusing to me. Is this just showing that the proposed algorithm succeeds in reducing LTP after 40 gradient steps? Why don't we compare the mean / variance of returns for complete training runs that do or don't use rejection? The 'x' markers show points reached by full training runs, so why can't we show that with/without the proposed approach? Or perhaps I'm misunderstanding, and Algorithm 1 is meant to by applied after a training run in order to robustify an already learned policy?

I felt the experiments could be more comprehensive w.r.t environment diversity. Given that the main claims of the paper are substantiated only empirically, I think it is potentially a problem that the paper only considers a few DMC environments (ant, walker2d, halfcheetah, hopper) which all involve locomotion.

**Questions:**

"For our analysis, we use 200 policies generated by different runs of TD3. From these we select pairs of policies with different post-update return distributions, as measured by their standard deviation or left-tail probability, but similar mean" (L214-216).
I don't fully understand how pairs of policies were selected using these criteria. Were these handpicked? Is there a more systematic way of picking pairs of policies? Is it possible for us to consider more than (a handful+1) examples?

What is the claimed contribution of Algorithm 1? Are the authors claiming that this update scheme should be used practically? Is this just an illustration due to other considerations (e.g. computational cost).

I think you should probably mention distributional RL [1] as related work.

In Figure 3, labeling the successful/failing trajectory either in the plot or in the caption would be nice.

===
I have read the rebuttal and bumped up my score slightly in light of the additional experimental results.

---

> ### Author Rebuttal · Authors · 2023-08-10
>
> Thank you for your feedback!
>
> > I suspect there is some way to quantify the claims in this section across a larger population of policy pairs, which I think would significantly strengthen this section.
>
> We thank the reviewer for the suggestion. To quantify whether there is a statistically significant difference in the proportion of return collapses encountered when interpolating between policies, we use the following experimental design. We sample for each environment a set of 500 pairs of policies from the same runs and a set of 500 pairs of policies from different runs. Then, we linearly interpolate between policies in the pairs, producing 100 intermediate policies, and randomly perturb them using Gaussian noise with standard deviation $0.0003$ to obtain an estimate of the mean of their (random) post-update return distribution. Then, for each pair of policy, we measure how frequently the return collapses in between the two extremes, by counting how many times it becomes less than 10\% of the minimum return of the two original policies. We then average this _Below-Threshold Proportion_ across pairs, and across environments using rliable (Agarwal et al., 2021). Figure 1 (a) shows a proper quantification of the phenomenon: on Brax, there is on average almost no drop in return when interpolating among policies from the same run. Additionally, similar results on Atari games in Figure 1 (b) (see response to ZEGo) show that the phenomenon also exists in other domains.
>
>
> > I felt the experiments could be more comprehensive w.r.t environment diversity.
>
> We provide additional results in the pdf using a set of games from the ALE (see General Response). We also have additional results on DeepMind Control Suite in the Appendix.
>
>
> > Or perhaps I'm misunderstanding, and Algorithm 1 is meant to by applied after a training run in order to robustify an already learned policy?
>
> This is correct -- the purpose of Algorithm 1 is to transport an existing good policy to a less noisy neighborhood of parameter space. Training an agent from scratch with the procedure in Algorithm 1 would slow down progress, especially at the beginning of training. The pipeline we study involves achieving a 'good' policy (with respect to its return) using existing RL techniques, and then 'fine-tuning' the resulting policy with Algorithm 1.
>
> > I think you should probably mention distributional RL [1] as related work.
>
> Thanks for the suggestion, we will add that to the related work.
>
> > In Figure 3, labeling the successful/failing trajectory either in the plot or in the caption would be nice.
>
> Thank you for the suggestion, we will add this to the final version of the paper.

---

> > ### Comment · Reviewer_Voa2 · 2023-08-12
> >
> > Okay I think most of my concerns have been reasonably addressed, specifically w.r.t more thorough experimental evaluation and more rigorous statistical analysis for Section 4.1. I am still in favor of acceptance and bumped my score from a 6 to a 7.

---

### Official Review · Reviewer_QuAp · 2023-07-06

**Soundness:** 3 good
**Presentation:** 3 good
**Contribution:** 2 fair
**Rating:** 5
**Confidence:** 3

**Summary:**

In this study, the authors explore the underlying causes of the instability observed in the performance of deep RL policies. Their investigation revolves around examining the relationship between a policy and its corresponding return, with a focus on analyzing how minor variations in policy parameters impact the overall return. The authors posit that regions where even slight perturbations in policy parameters result in significant changes in return are susceptible to failures. To address this issue, they introduce an algorithm designed to navigate away from these volatile neighborhoods, thereby enhancing the robustness of the policy. Lastly, the authors provide a set of experimental results to demonstrate their findings and validate the efficacy of their optimization algorithm through another set of experimental results.

**Strengths:**

Overall Strengths:
1. The organization and writing of this paper are of good quality. The authors have succeeded in presenting the content in a clear and easily understandable manner, which greatly enhances the readability of the paper. A notable aspect is the structure of the experiments sections (Section 3 and Section 4), where the authors first introduce the research questions, followed by a detailed explanation of the experimental setup, and finally present the results along with their corresponding conclusions. This approach significantly aids in the comprehension of the experiments, in such heavily experiments-based paper.
2. I personally find the high-level idea of studying why the performance of RL policies so unstable well-worth pursuing.


**Weaknesses:**

1. This paper would benefit from providing additional explanation and motivation regarding the significance of investigating the mapping between policy parameters and return. A more thorough exploration of why this particular landscape is crucial to the focus of the work is needed. Expanding on the importance of understanding this relationship would enhance the reader's understanding and emphasize the relevance of the research.
2. The experimental setup is limited. I would personally be interested in how different algorithms compare with each other when trained on the same task as well as how a certain algorithm behaves when trained on different tasks.


**Questions:**

1. What will the results look like among policies trained with different algorithms on the same task?
2. I have seen recent works studying how the mapping between the state-space and the policy look like and how this landscape evolves during training. How do those findings relate to the findings in this paper? (For reference, look at “Understanding the Evolution of Linear Regions in Deep Reinforcement Learning” from NeurIPS’22)
3. I personally find figure 2 hard to understand. But for now, I don’t have any particular suggestions to improve the readability of these.
4. It would be invaluable if the same experiments would be repeated for policies trained with behavior cloning. To first see how policies trained with behavior cloning on some task compare to policies trained with other RL algorithms on the same task. And second, see whether different runs of behavior cloning demonstrate non-noisy return landscapes.


**Limitations:**

Please refer to the previous sections.

---

> ### Author Rebuttal · Authors · 2023-08-10
>
> Thank you for your feedback!
>
> > This paper would benefit from providing additional explanation and motivation regarding the significance of investigating the mapping between policy parameters and return. A more thorough exploration of why this particular landscape is crucial to the focus of the work is needed.
>
>
> By investigating the return landscape with the set of tools we introduced in the paper, we can both advance our understanding of the policies traversed by policy optimization algorithms and improve their stability. Our work offers both a new perspective on previously-studied unstable learning dynamics of deep RL agents (e.g. Chan et al. 2020, Henderson et al. 2018), as well as a new way to look at the stability of a given policy in terms of its post-update return distribution, which we measure using statistics such as the left-tail probability and show to correspond to the qualitative robustness an agent's behavior. Through this landscape-oriented view, we can improve a policy along that dimension of policy quality, as we propose to do in Algorithm 1. We are happy to make revisions to the manuscript to emphasize the central role of the return landscape in the results obtained.
>
> > How do those findings relate to the findings in this paper? (For reference, look at “Understanding the Evolution of Linear Regions in Deep Reinforcement Learning” from NeurIPS’22)
>
> We appreciate the reference. Although this work is similar in spirit to ours, in that it aims to characterize policies beyond common evaluation metrics, they study the complexity of the learned policy as a function of the input state, whereas we study the return in the environment as a function of the policy parameters. Naturally, these objects are related: we conjecture that a more complex policy may exhibit greater variability in returns under a single update to its parameters. We take this as an interesting direction for future work, and will cite this work in the final version of the paper.
>
> > I personally find figure 2 hard to understand.
>
>  To clarify, Figure 2 considers several main objects. Each color in the legend corresponds to an algorithm, and each point corresponds to a policy produced by that algorithm. For each policy, we compute its post-update return distribution. On the three main scatter plots, we plot each policy according to different statistics of the distribution: skewness, standard deviation, and left-tail probability, plotted against the mean. Therefore, each policy appears once on each scatter plot. We select 6 policies of interest, indicated by stars, and plot histograms of their post-update distributions at bottom to make a correspondence between the statistics and visual appearance of the distribution. We will consider if it's possible to simplify the presentation of Figure 2 for the final paper.
>
> > What will the results look like among policies trained with different algorithms on the same task?
>
> Figure 2 depicts precisely how policies obtained by different algorithms trained on the Brax `ant` task compare to each other. Similar figures are reproduced for other environments across Brax and DeepMind Control in the appendix.
>
> > It would be invaluable if the same experiments would be repeated for policies trained with behavior cloning
>
> We thank the reviewer for the compelling suggestion. We were especially curious to understand whether behavior cloning produces policies which occupy less noisy neighborhoods of the return landscape. To do so, we conducted a set of additional experiments on 4 Brax environments. The protocol was as follows: For each environment, we consider 10 independent training runs of TD3, and 5 policies distributed evenly throughout the run. For each of these policies, we train a new agent using behavior cloning on the data logged up until the collection time of the teacher policy, replacing the actions in the dataset with the actions of the teacher policy, for 1 million gradient steps. We log 10 policies throughout each training run of behavior cloning. To compute the post-update return distribution for the policies obtained by behavior cloning, we used one additional gradient step on the MSE-based BC objective, and 1000 samples.
>
> In Figure 3 (left), we compare statistics of the post-update return distributions for all pairs of policies $(\pi^{BC}_{i, j}, \pi^{TD3}_i)$ where policy $\pi^{BC}_{i, j}$ is obstained by behavior cloning $\pi^{TD3}_i$. We compute the Pearson correlation coefficient of statistics of the post-update return distributions of these policies: between the mean of each pair, and between the LTPs of each pair. We find that the means are highly correlated and that the learned BC policies are comparable in performance to their teacher policy. We additionally show that correlation in the LTP is much more variable across environments -- in general, cloning a policy of high or low LTP does not always lead to a cloned policy of the same LTP.
>
> But does training policies by BC produce policies occupying less noisy neighborhoods of the return landscape, which would have correspondingly lower LTP overall? In Figure 3 (right), we show the mean LTP across policies and environments, along with its 95% bootstrapped confidence interval following the recommendations from *rliable* (Agarwal et al, 2021). Our results demonstrate that BC does not produce fundamentally more stable policies, as measured by the LTP.

---

> > ### Comment · Reviewer_QuAp · 2023-08-14
> >
> > I appreciate the authors' thorough response and their effort in addressing my queries.  I especially find the new behavior cloning results very interesting.
> >
> > After a careful re-evaluation of this work, looking at the insights from fellow reviewers, and considering the authros' efforts to address reviewers' questions and suggestions, I have decided to increase my score as I believe this work holds potential value for the broader RL community.

---

### Official Review · Reviewer_ErLG · 2023-07-07

**Soundness:** 2 fair
**Presentation:** 1 poor
**Contribution:** 2 fair
**Rating:** 4
**Confidence:** 4

**Summary:**

This paper looks at visualizing the return landscape in typical continuous control environments of deep RL. The authors propose a statistic over the returns profile, namely the left tail probability, and try to present this as a suitable measure of policy quality in a series of experiments and visualizations. Finally, they propose a high-level rejection sampling process for the policy updates using the proposed statistic and argue that it can lead to similar-mean but less-variable return profiles in the trained policies.

**Strengths:**

The idea of looking at return surfaces is not original (e.g., see [1], [2], and [3] among many others). However, trying to propose a remedy based upon such visualizations has been less common. In terms of quality, the paper tries to narrow down its focus to control and robotics applications, which is admirable. As for the clarity of the paper, too much theoretical complications did not hinder the point of the paper. Unfortunately, the results are less significant than expected, but the studied topic (i.e., better understanding of the return landscape for deep RL in control environments) is certainly of great importance and has been under-looked for too long.

# References
[1] Sullivan, R., Terry, J. K., Black, B., & Dickerson, J. P. Cliff Diving: Exploring Reward Surfaces in Reinforcement Learning Environments. ICML, 2022.

[2] Bekci, Recep Yusuf, and Mehmet Gümüş. "Visualizing the Loss Landscape of Actor Critic Methods with Applications in Inventory Optimization." arXiv preprint arXiv:2009.02391 (2020).

[3] Ilyas, A., Engstrom, L., Santurkar, S., Tsipras, D., Janoos, F., Rudolph, L., & Madry, A. (2018). A closer look at deep policy gradients. ICLR 2020.

[4] Hwangbo, Jemin, et al. "Learning agile and dynamic motor skills for legged robots." Science Robotics 4.26 (2019): eaau5872.

[5] Saleh, E., Ghaffari, S., Bretl, T., & West, M. (2022). Truly Deterministic Policy Optimization. Advances in Neural Information Processing Systems, 35, 8469-8482.

**Weaknesses:**

## Presentation and Writing
First and for most, the paper lacks a clear set of contributions. I encourage the authors to dedicate the last paragraph of their introduction to list 3 or 4 main contributions, and commit to delivering them in the entire paper. This is a standard practice, and lacking this makes it extremely difficult to evaluate the merit of the work.

Second of all, the paper lacks a main message and a proper effort to convince the reader about the validity and the importance of it. As is, the manuscript looks more like a technical report draft rather than a conference paper. In particular, note that most NeurIPS papers focus on driving the main message in the initial sections, and avoid detailing the methods/data until the experiments section. However, this paper lacks such a structure, and tries to narrate a story by jumping from one experiment to another, while losing a cohesive context and spending a lot of space on detailing experimental details.

## Rigor
The paper is not driven by any amount or kind of theory. Instead, the authors choose or abuse vague terms and intuitions to describe a series of hypotheses. For example, the notion of "stability" of a policy gets thrown around a lot in the paper (e.g., Line 246) without any mathematical standing or definition. This is despite the fact that the stability of a policy is a well-defined topic of research, and has many measures such as the Lyaponuv stability criterion. I doubt the authors were targeting this original notion of stability, and this is just one example of a technical term being abused.

Many other vague terms are consistently used throughout the paper, including, but not limited to, (1) the "quality" of a policy, (2) "noisy" neighborhoods, (3) "failure" of a policy or a trajectory, (4) "safer" behavior, and so on.

## Premise
  1. Most deep Policy Gradient methods (e.g., PPO, TRPO, DDPG, TD3, SAC, etc.) optimize the mean of the discounted return in the MDP framework. Other statistics of the return distribution (e.g., the variance, the mode, left tail probability, etc.) are irrelevant to the optimization objective. When evaluating methods and policies, it is standard that higher mean payoffs are preferable even if the return distribution has higher variance.

  If a particular application requires consistent returns from a policy, the user either has to (1) engineer the reward so that such variations lead to worse mean returns, or (2) use PG methods that optimize a different objective (e.g., the worst-case performance with min-max optimization methods). Both approaches are commonly used in control and robotics applications.

  Of course, consistent policy performance is preferable but it shouldn't be confused with the main performance metric. The entire paper is written on the premise that variations in the return are inherently undesirable, and I don't necessarily agree with that. In my experience, Deep RL algorithms such as PPO, SAC, and TD3 gravitate towards such "noisy" regions at the end of the training because these regions can potentially yield higher returns than the stable areas, even if the return improvement is small.

  When evaluating such PG methods, only the mean return should be considered. The only way the paper's findings could be relevant is if high-variations in the return profile were shown to correlate with lower mean-returns at the end of training. I'm afraid the paper is not showing that.

  2. Deep PG methods do not directly optimize deterministic policies. Each algorithm, even the deterministic ones such as DDPG and TD3, have an exploration parameter $\sigma$. These parameters cause a filtering/convolutional effect on the return landscape, and smoothen the effective optimization objective. The paper mainly focuses on oscillations in the deterministic landscape, whereas the effectively optimized landscape can be much smoother and I'm not sure the same findings are applicable to the filtered landscape.

## Arguments

The experiments and figures in the paper aren't always convincing or helpful. In particular,

  1. Figure 5 doesn't prove a meaningful hypothesis. Such parameter interpolation behavior may not even be specific to RL problems, and "non-noisy" problems can also exhibit the same behavior. In other words, interpolating parameters within the same run in a deep classification task can lead to reasonably-performing parameters while inter-run interpolations wouldn't perform well. I'm not sure how this observation is a solid evidence for the existence of special patterns or connections, even if there were many gradient steps in between interpolated parameters.

  2. Figure 1 consumes a lot of space, but carries little information.

  3. Also, the paper spends a lot of time speculating. For instance, Lines 49-52, Lines 236-243, and Lines 202-207 are a few examples. Line 245 speculates about the existence of "paths" (I'm not even sure what it means), and such claims about high-dimensional optimization problems cannot be easily proven by the experiments conducted in this paper.

**Questions:**

I would like the authors to address my concerns in the weakness section regarding the premise of the paper and the made arguments.

**Limitations:**

Given the the recent reproducibility issues with deep RL, papers in this are must be held to a higher standard. In particular, for this paper:
  1. Many depicted training curves are based upon single trainings.
  2. 20 seeds were used for running the experiments. I encourage the authors to run the experiments for at least a 100 seeds.
  3. Most figures lack any information about the uncertainty of the statistics. Finding ways to incorporate confidence intervals in the plots is important for this paper.
  4. The number of environments considered in the paper are limited. Hopper, Walker, Half-Cheetah, and Ant were shown in the main paper, while there are many more environments in the standard gym benchmarks.
  5. The quality of the environments in the paper could also be improved. In particular, while standard gym benchmarks are a good start, they do not depict realistic robotic artifacts. There are many realistically modeled robotic environments, such as the ones used in [4] and [5], that can be better representatives for validation of findings.
  6. While the paper is specifically targeting continuous control, the authors never discuss the physical aspects of each environment and do not make physical variations to the underlying physical models.
  7. The code is not shared open-source in the paper. While this is not a requirement for the conference, it is certainly of great important to the RL community. I encourage the authors to release an open-source link to the code in the main paper.

---

> ### Author Rebuttal · Authors · 2023-08-10
>
> Thank you for your feedback!
>
> > The idea of looking at return surfaces is not original (e.g., see [1], [2], and [3] among many others).
>
> While the return landscape has been studied in other works, our distributional perspective on it has no precedent. It is also important to distinguish between works which study a loss landscape (e.g. [2]), and a landscape of returns from the environment, which we study.
>
> Of the works in the latter category, [1] emphasizes visualizations of the landscape and [3] studies its relationship with the optimization landscape. Neither of these works show how distributional properties of the landscape can characterize policies, nor do they demonstrate macro-scale structure of the landscape when interpolating between policies.
>
> > I encourage the authors to dedicate the last paragraph of their introduction to list 3 or 4 main contributions, and commit to delivering them in the entire paper.
>
> Our introduction already explicitly states our contributions, in the form of three paragraphs with bold title, each one corresponding to a specific contribution.
>
> >  the manuscript looks more like a technical report draft rather than a conference paper.
>
> >  most NeurIPS papers focus on driving the main message in the initial sections, and avoid detailing the methods/data until the experiments section.
>
> We do not present any experimental detail in the introduction, and simply enumerate and explain our findings. Note that the overall presentation strategy has been praised by most of the other reviewers.
>
>
> > vague terms are consistently used throughout the paper, including, but not limited to, (1) the "quality" of a policy, (2) "noisy" neighborhoods, (3) "failure" of a policy or a trajectory, (4) "safer" behavior
>
> These concepts are defined in the paper or in related work. Notions such as "stability" and "failure" are entailed by the post-update return distributions and the LTP. "Noisy neighborhood" is a name defined in line 31. "Safer" is used in the sense reported in the related work. Better "quality" means having lower level of LTP/std for a given level of mean return of a post-update return distribution.
>
> > variations in the return are inherently undesirable, and I don't necessarily agree with that. In my experience, Deep RL algorithms such as PPO, SAC, and TD3 gravitate towards such "noisy" regions
>
> An important point behind our analysis based on post-update return distributions is showing that a policy optimization algorithm produces policies with the same level of return but very different levels of variability under further updates. Our experiments have the goal of quantifying and understanding this phenomenon, which is generally undesirable when deploying a policy, but also compelling from a purely scientific perspective.
>
> > The paper mainly focuses on oscillations in the deterministic landscape, whereas the effectively optimized landscape can be much smoother
>
> Our paper is about the return landscape resulting from evaluating a policy in the environment, regardless of how smooth the optimization objective used by the algorithm that produced this policy was.
>
> > interpolating parameters within the same run in a deep classification task can lead to reasonably-performing parameters
>
> The reviewer is correct that a similar phenomenon exists in deep classification (see citations). However, the RL setting is different: the optimization objective is non-stationary and the evaluation metric (the return vs the loss) depends on an environment and multiple forward passes from the neural network. We believe this makes the existence in RL of a phenomenon akin to mode connectivity far from certain.
>
> > Line 245 speculates about the existence of "paths" (I'm not even sure what it means)
>
> A path in this context simply refers to a trajectory in parameter space. Thus, a linear path is one that traverses through parameter space along a line, which is given by the interpolation between two parameters.
>
> > Many depicted training curves are based upon single trainings.
>
> Please note that our paper does not report any single training curve, since they are not the object of study of our work.
>
> > 20 seeds were used for running the experiments. I encourage the authors to run the experiments for at least a 100 seeds.
>
> We disagree that additional seeds are required to support the results presented in the paper. Using 100 seeds instead of 20 would only cause the scatter plot in Figure 2 to be populated by more points, and not increase the significance on any of the new plots in the pdf, greatly increasing the computational cost of our experiments without any clear benefit.
>
> > Finding ways to incorporate confidence intervals in the plots is important for this paper.
>
> We incorporated error bars in experiments in the pdf using bootstrapped CIs (as in Agarwal et al., 2021). First, we compute the uncertainty of the statistics of the post-update return distribution (see Figure 6), showing it is very small; second, we quantify the presence of return drops when interpolating across pairs of policies (see Figure 1 and response to Voa2); third, we show the improvement provided by Algorithm 1 across environments is statistically significant (Figure 8); fourth, we measured the number of rejections from Algorithm 1 (see Figure 7).
>
> > The number of environments considered in the paper are limited.
>
> See Appendix and general response.
>
> > while standard gym benchmarks are a good start, they do not depict realistic robotic artifacts.
>
> > the authors never discuss the physical aspects of each environment
>
> Our paper is mainly directed at the deep RL community. Since simple robotic locomotion tasks or Atari games are standard in this community, we believe the extension of our analysis to more complex robotic artifacts is an interesting avenue for future research, but outside the scope of our paper.
>
> > I encourage the authors to release an open-source link to the code
>
> We will release code once the paper is published.

---

> > ### Comment · Reviewer_ErLG · 2023-08-17
> > **Response to Rebuttals**
> >
> > I thank the authors for their response, and I’ve raised my score. I found the additional experiments were very valuable.
> >
> > That being said, I still find it difficult to recommend this paper for acceptance. The following are just some of the reasons:
> >
> > 1. The authors mostly did not commit to making changes to the paper in the rebuttals. Instead of isolating the response to the reviewer, I wished to note the specific changes & improvements the authors would make to the paper itself. The wealth of questions here indicate interest in the topic, but also show that the writing and the clarity of the paper could be improved.
> >
> > 1. The nomenclature problem is still persistent:
> >     * The parameter to return function is mostly deterministic in this paper. The “noisy” naming implies stochasticity in the returns, which is not the main feature of the study. Note that “chaotic” functions, where small input variations can lead to large variations in the output, are very different from “noisy” functions.
> >     * The response “these concepts are defined in the paper or in related work” is inadequate for the justifying the misuse of such an established notion as “stability”. Stability in continuous control has been a major topic of research for the better part of a century, and there are tens/hundreds of papers in NeurIPS alone on the notion of stability in non-linear control systems. Any traditional control textbook from 50 years ago have already used the same term with an entirely different meaning. To be clear, recent works such as Reference 21 (Nikishin et. al in the UAI workshop of 2018) did not justify this naming as well.
> >      * “Safety” in reinforcement learning is a separate topic of research that this paper is not about, and this word also must be replaced with an alternative and a more proper notion.
> >      * “Paths” may better be replaced by linear interpolations throughout the paper; the former is just too general and vague.
> >
> > 2. Dismissing the connection between the optimization objective and the return landscape, on the basis of a limited scope or irrelevance to the work, is not a valid argument in my opinion. The two are significantly connected to each other, and they must be studied and motivated in conjunction with each other. This is a glaring shortcoming with the paper as it is, and I tried to make this point as clear as possible in my earlier review.
> >
> > 3. There are 12 plots in Figure 5, and each one is produced by picking one or two trainings, sampling two policies from those training, and then interpolating between the policies. The small number of trainings and environments involved in this process make this evidence merely anecdotal. Adding more plots is also not a scalable approach to make this experiment more comprehensive. The authors should have found better ways of visualization to present a more statistically reliable evidence.
> >
> > 4. The authors mention that running more seeds can only make the plots more crowded. This, on itself, should have been an alarming bell with the experimental design in this case. To contrast this with an example, most papers report the mean statistics with a confidence interval which only gets smaller and more reliable with more seeds. The authors could have found better statistics and methods of visualization that could scale to many more seeds and environments without sacrificing any clarity.
> >
> > 5. I hope the authors don’t confuse the “initial sections” with the “introduction”. In general, Lines 126-140 could have been out-sourced to the appendix, and Figure 1 is using too much space without presenting a lot of valuable information. Similarly, I don’t find Figure 4 necessary in the main text, as those failure visualizations are too known for deep RL practitioners and roboticists. Instead, more important and missing experiments could have been included in the main body.

---

> > > ### Author Response · Authors · 2023-08-17
> > >
> > > Thank you for your response!
> > >
> > >  1. We committed to make some changes to our paper in the rebuttals. In particular, we will include all the figures contained in the rebuttal pdf in the final version of the paper. The rest of this answer also includes additional modifications that will be making to the paper.
> > > 2. We appreciate your point on nomenclature, but largely disagree with it.
> > >     - The "noisy" term can be indeed seen as being associated with the stochasticity of returns from the post-update return distribution; even if the return is actually a function, a defining feature of our study is the use of distributional tools to characterize it. Therefore, we believe that "noisy neighborhood" has an appropriate degree of accuracy.
> > >     - On "stability", we will make more clear in the paper that the concept of stability we refer to is the one associated to the Left-Tail Probabilility of the post-update return distribution of a policy. Note that related notions of stability have been employed in other work, beyond (Nikishin et al., 2018), for instance as "variability within training runs" in (Chan et al, 2019) or in (Khanna et al, 2022).
> > >     - To avoid confusion, we will only keep the term "safety" to refer to previous work and substitute any reference to it with "stability" in the LTP sense.
> > >     - The word "paths" has been largely used in previous work on mode connectivity, to denote a sequence of vectors that is connected in a parameter space, for instance by a linear interpolation. To provide concrete examples, (Frankle et al, 2019) writes "networks are connected by linear paths of constant test error" and the recent Git Re-Basin paper (Ainsworth et al., 2022) writes "connected by paths of near-constant loss".
> > > 3. We apologize for the confusion, due to the lack of space in the original rebuttal. The connection between the optimization objective and the return landscape is indeed at the heart of our work. The post-update return distribution is the bridge between optimization, by the execution of multiple updates, and returns, by evaluation of the resulting policies, even in the absence of explicit visualization of the optimization landscape.
> > > 4.  We indeed already provided them through Figure 1 in the rebuttal pdf, and the response to reviewer Voa2, where we describe the protocol we used to quantify our findings on interpolation between policies. In general, our results demonstrate large and statistically significant differences in the quantity of collapses in return when interpolating between policies from the same vs. different runs, across environments. We believe that the inclusion of some anecdotal results has illustrative value, but we agree that the conclusions here are best supported by rigorous statistical comparisons.
> > > 5. While we agree that aggregate metrics are a commonly-used tool for making comparisons, we strongly believe that results showing the behavior of individual policies or training runs can be valuable.
> > >
> > >     To be precise, in our context, we stated that more seeds would "cause the scatter plot in Figure 2 to be populated by more points." This figure, in particular, is an intentionally disaggregated result which illustrates that the post-update return distributions of policies obtained by three popular algorithms exhibit diverse profiles and statistics. We hold that the 20 seeds per algorithm shown are enough to support this claim.
> > >
> > >     During the rebuttal period, we presented several new results aggregated over multiple seeds and environments, along with appropriate confidence intervals. These include: first, providing CIs for the statistics of the post-update return distribution (see pdf Figure 6), showing they are very small; second, quantifying the presence of return drops when interpolating across pairs of policies (see pdf Figure 1 and response to Voa2); third, demonstrating the benefit provided by Algorithm 1 across environments is statistically significant (see pdf Figure 8); fourth, measuring the proportion of rejections from Algorithm 1 (see pdf Figure 7). In all cases, the computed confidence intervals demonstrate that we have used enough independent runs to provide statistically significant comparisons, refuting the proposition that additional seeds would benefit our work.
> > >
> > > 6. We acknowledge the reviewer's suggestion. In order to accommodate the several new results from the rebuttal, we will condense lines 126-140, decrease the size of Figure 1 and, if needed, remove Figure 4.

---

> > > > ### Comment · Reviewer_ErLG · 2023-08-19
> > > > **Third Reviewer Response**
> > > >
> > > > I thank the authors for their response.
> > > >
> > > > * **Comment**: *The "noisy" term can be indeed seen as being associated with the stochasticity of returns from the post-update return distribution; even if the return is actually a function, a defining feature of our study is the use of distributional tools to characterize it. Therefore, we believe that "noisy neighborhood" has an appropriate degree of accuracy.*
> > > >
> > > >    **Response**: I'm still not convinced about the validity of the "noisy landscape" naming scheme. By this logic, any function can still define a "noisy landscape" as long as a stochastic-enough optimizer is applied to them; looks like the "noisy landscape" is a property of the parameter update process and not the function itself. Unfortunately, I don't find that "noisy neighborhood" has an appropriate degree of accuracy. I will leave it to the authors to propose a proper alternative. This is specially important as this area is new and growing.
> > > >
> > > > * **Comment**: *We apologize for the confusion, due to the lack of space in the original rebuttal. The connection between the optimization objective and the return landscape is indeed at the heart of our work. The post-update return distribution is the bridge between optimization, by the execution of multiple updates, and returns, by evaluation of the resulting policies, even in the absence of explicit visualization of the optimization landscape.*
> > > >
> > > >    **Response**: I agree with the authors, but my point was not that. In particular, no specific optimization algorithms or processes were studied in conjunction with the paper's observations about the post-update return. The paper, as it is, is referring to "post-update" returns collectively for all RL algorithms, and the number of methods studied in the paper isn't even enough to properly support such generalizations. To help the reader understand what is specifically at the root of these observations, the paper must have included some studies on which specific aspects of the existing Deep RL algorithms is causing the phenomena. Again, this is a major shortcoming of the paper, and I find it difficult to accept that this should be left to future work or that it is reasonably beyond the scope of this work.
> > > >
> > > > * **Comment**: *We believe that the inclusion of some anecdotal results has illustrative value, but we agree that the conclusions here are best supported by rigorous statistical comparisons.*
> > > >
> > > >    **Response**: I thank the authors, and agree that the conclusions here should be supported by rigorous statistical comparisons. This point, specifically, should be clearly mentioned in the main paper's limitations.
> > > >
> > > > * **Comment**: *In all cases, the computed confidence intervals demonstrate that we have used enough independent runs to provide statistically significant comparisons, refuting the proposition that additional seeds would benefit our work.*
> > > >
> > > >    **Response**: I thank the authors for their experiments. However, the authors should know that with **5 PPO runs**, even the confidence intervals are too unreliable to comfortably claim statistical significance.
> > > >
> > > > * **Comment**:  *Note that related notions of stability have been employed in other work, beyond (Nikishin et al., 2018), for instance as "variability within training runs" in (Chan et al, 2019) or in (Khanna et al, 2022).*
> > > >
> > > >     **Response**: As I already stated, the literature on stability goes back longer than a century, and clearly precedes any referenced work. In fact, I specifically mentioned (Nikishin et al., 2018) in my original response to address this head on, and avoid having a citation match.
> > > >
> > > > To be abundantly clear, I stand by my original review and I find the following changes necessary before I can recommend this work for acceptance.
> > > > 1. Propose an alternative and proper naming to "noisy landscapes" throughout the paper and commit to replacing it.
> > > > 2. Propose an alternative to the "stability" notion, and commit to replacing it throughout the paper.
> > > > 3. Clearly dedicate one or two lines in the limitations to the fact that "the conclusions here should be supported by rigorous statistical comparisons in future work".
> > > > 4. Apply all of the other changes the authors promised. Also, the rebuttal experiments should be repeated with at least the same level of statistical quality as the original paper (e.g., no less seeds than 20). I understand the computational and time constraints within the rebuttal period, but almost all of the rebuttal experiments should have already been part of the original paper in my humble opinion and the authors will have a plethora of time before the final revision deadlines to repeat the experiments with a more reasonable quality.
> > > >
> > > > Unfortunately, I won't have more time to respond back, and I don't think further debate will be helpful. If the authors can commit to making *all of the aforementioned changes*, I will be happy to recommend this work for acceptance. Otherwise, I will maintain my current score as is.

---

> > > > > ### Author Response · Authors · 2023-08-20
> > > > >
> > > > > We are glad to hear the suggestions from the reviewer. However, we disagree that every aspect of the proposed changes would be useful for improving the paper. To respond to the reviewer's points:
> > > > >
> > > > > - We still believe the use of the expression "noisy neighborhood" is appropriate for scientific communication reasons. The term is related to stochasticity as already discussed, and overall it is just a proper name we have chosen.
> > > > > - When its meaning is ambiguous, we will substitute the term "stability" with appropriate expressions such as "post-update return variability".
> > > > > - We will highlight more the limitations of the work in terms of comparisons, where applicable (e.g., it would be interesting to know if our conclusions generalize to more environments and algorithms). However, note that our sentence "our conclusions here are best supported by rigorous statistical comparisons" referred to the statistical comparisons we actually already carried out for the rebuttal, and that can be found in the rebuttal pdf. This might not be clear if the sentence is extrapolated from its context.
> > > > > - As stated in the previous response, we will apply the other changes and experiments added during the rebuttal period to the paper. We will also run more seeds for the ALE experiments to align them with the Brax/MuJoCo ones.
> > > > >
> > > > > We hope these comments clarify our position on the issues highlighted by the reviewer.

---

> ### Comment · Area_Chair_koBB · 2023-08-16
> **Are you satisfied by the answers?**
>
> Dear reviewer,
>
> Would you please indicate whether the authors' response is satisfactory for you? If not, please engage with the authors, so we can get a better assessment of this work.
>
> Thank you,
> Area Chair

---

### Official Review · Reviewer_ZEGo · 2023-07-18

**Soundness:** 3 good
**Presentation:** 3 good
**Contribution:** 4 excellent
**Rating:** 7
**Confidence:** 4

**Summary:**

This work proposes a new perspective for investigating the stability of RL algorithms -- the return distribution of post-update policies. A simple rejection technique is developed correspondingly to increase the stability in that sense.

**Strengths:**

The paper is very well written and easy to follow. This new perspective of return distribution of post-update policies appears novel and makes much sense. I can imagine many followup directions of this work. The rejection technique is neat while effective.

**Weaknesses:**

There is indeed a few weaknesses in the treatment of the problem.
1. Why do the authors limit the investigation to deterministic transitions and continuous actions? In my understanding the concept the authors propose apply to stochastic environments and discrete actions as well.
2. Maybe it's better to rename the work as "Policy Optimization in a Noisy Neighborhood: On Return Landscapes in MuJoCo" since the work in its current form does not have any other domains. In other words, it is interesting to see whether the observation is general across various domains, especially the stability of interpolation of policies inside a single run.

That being said, I believe the novelty of this work is significant and am happy to increase my score based on the authors' response to my concern.

**Questions:**

see above

**Limitations:**

see above

---

> ### Author Rebuttal · Authors · 2023-08-10
>
> Thank you for your feedback!
>
> > In my understanding the concept the authors propose apply to stochastic environments and discrete actions as well.
>
> We thank the reviewer for the suggestion. We have run two experiments (Brax with stochastic policies and four games from the ALE, see General Response) to confirm that this is the case.
>
> For the stochastic setting, we generalize our experimental protocol for the computation of the post-update return distribution to the case in which the policy is non-deterministic. For each policy checkpoint from a TD3 training run, we produce 100 policies with the TD3 update, and then evaluate the stochasticity policy obtained with the same random perturbation scale used during training 10 times and compute the resulting distribution statistics. We find that, for a given policy, this alternate post-update return distribution yields very similar results to the one based on its deterministic version (see Figure 2 in the pdf). In the paper, we focus on deterministic environments in order to understand the relationship between policy parameters and return without the confounding effect of environmental stochasticity.
>
>
> > Maybe it's better to rename the work as "Policy Optimization in a Noisy Neighborhood: On Return Landscapes in MuJoCo" since the work in its current form does not have any other domains.
>
> We appreciate the point from the reviewer. Including the results in the Appendix and the new results on the discrete-action ALE, we have now a total of three different simulators (Brax, MuJoCo via DMC, and Atari). We believe this positions the method of analysis to be general, despite, for computational reasons, we have run most of our experiments on the efficient Brax simulator.
>
> > In other words, it is interesting to see whether the observation is general across various domains, especially the stability of interpolation of policies inside a single run.
>
> We thank the reviewer for the suggestion. To test whether the phenomenon is present also on other domains, we run an interpolation experiment on the ALE under the same setting based on PPO described in the general response. As per suggestion of other reviewers, we produce quantitative results on the phenomenon with the following procedure. For each one of the four games, we sample a set of at least 20 pairs of policies from same runs and 20 pairs of policies from different runs of the algorithm. Then, we linearly interpolate between policies in the pairs, producing 50 intermediate policies, and randomly perturb them using Gaussian noise with standard deviation $0.0003$ to obtain an estimate of the mean of their (random) post-update return distribution. Then, for each pair of policy, we measure how frequently the return collapses in between the two extremes, by counting how many times it becomes less than 10\% of the minimum return of the two original policies. We then average this _Below-Threshold Proportion_ across pairs, and across environments using rliable (Agarwal et al., 2021). Figure 1 (b) shows that the phenomenon, properly quantified, is still present when using a very different class of environments (discrete-action, game-based).

---

> > ### Comment · Reviewer_ZEGo · 2023-08-15
> > **Thanks for the additional result**
> >
> > I have increased the score from 6 to 7 in light of the author response.
> > I do encourage the author to include more ALE games in next revision -- apparently the authors have the required computational resource.

---

### Author Rebuttal · Authors · 2023-08-10

We thank the reviewers for their helpful comments and feedback. Overall, the reviewers found our perspective to be novel (ZEGo, Voa2, Lizq), our paper to be well-written (ZEGo, QuAp, Voa2, Lizq) and the quality of our figures and experiments to be high (Voa2, Lizq).


Multiple reviewers (ZEGo, ErLG, VoA2) were curious to know whether our method of analysis based on post-update return distributions and our results generalize to other domains. We provide an affirmative answer by showing that the tools we introduce are easily applicable to discrete-action environments (see Figures 4, 5, 1 (b) in the pdf), and that meaningful variations in the post-update return can be found across different settings.


As discrete-action environments, we employ four games from the standard ALE benchmark (Bellemare et al, 2013). We run PPO on these environments for 5 runs and for 10 million steps each, and measure post-update return distribution statistics using the same protocol employed in the rest of our paper: we evaluate 10 policies evenly-spaced across training, and perform 1000 independent updates to each policy. We find that post-update return distributions computed for these environments still exhibit a remarkable degree of variation, as captured by their standard deviation. At the same time, the shape of the resulting distributions can be quite different compared to the ones observed in robotic locomotion tasks, and, thus, it is not necessarily described in a rich way by metrics such as the LTP (see Figure 5).

Overall, these results reinforce the utility of return landscapes and post-update return distributions as a general tool to understand policies produced by deep reinforcement learning algorithms.

Reviewers also raised a range of individual issues, which we have addressed. This includes clarifying the motivation behind the paper, reporting more precise quantitative evidence for our findings, and clearing up some misunderstandings. We address these in replies to each reviewer below.

---

### Decision · Program_Chairs · 2023-09-21

**Decision:**

Accept (poster)

**Comment:**

The paper empirically studies the return landscape of RL algorithms through the lens of "noisy post-update neighbourhood". This refers to the return of a policy after an update of an RL algorithm. The noise is because of the stochasticity of the update given the mini-batch. The paper considers a few deep RL algorithms (SAC, PPO, TD3) and shows that the distribution of this post-update return affects how reliable the learned policy is, among several other related observations .

Most reviewers are positive about this paper. They believe that the paper provides a fresh perspective, is insightful, and the experiments support the main claims of the paper in overall.

There were some negative concerns about this paper, for example the limitation of the environments to MuJuco ones (which was partially addressed during the rebuttals by adding some ALE results), and several others concerns raised by Reviewer ErLG. For example, the reviewer has objections in how terms such as stability or safety have been used in the paper. Stability has a specific meaning in control theory, which is different from this paper's usage.

I read the paper myself. I believe that the paper offers new tools and empirical methodology to analyze RL algorithms. I also see that one may question certain aspects of the paper, including whether the measured quantities are actually what matters or the terminologies are the most accurate ones.

All considered, I believe what this paper offers clearly surpasses its shortcomings. As a result, I **recommend the acceptance** of this work and encourage the authors to consider the comments by the reviewers in order to improve their work.


I have some additional questions, which I hope the authors address in their next iteration of their paper:

- The paper focuses on the return of the post-update policies, showing that there are regions of the policy parameter space with high variations and heavy lower tail. What about the variations of the post-update parameters? Of course, this is a bit tricky to measure, as the parameters are a vector and not a scalar as return is, but I think some statistics of the parameter change, perhaps simply the norm, can be reported.

- In Figure 5, what are the (average) distance in theta_0 and theta_1 for two different cases of Same and Different runs? Are they comparable?
The dynamics of the training of a neural network is nonlinear, and it may converge to different basis of attractions for each run.

- The paper has a claim that "such a policy may correspond to a safer behavior, ... and possibly to deal with other unexpected sources of perturbation during deployment" (LL152-154). What kind of perturbations during the deployment are you thinking of? If the parameters of the policy are not corrupted, which is a safe assumption unless we lossy compress the neural network for deployment, the policy parameters do not change. Are the authors implying that having a smoother post-update neighbourhood somehow has robustness (in the sim2read context, for example) implications? If so, why?